# Low-coordinated copper facilitates the *CH$_2$CO affinity at enhanced rectifying interface of Cu/Cu$_2$O for efficient CO$_2$-to-multicarbon alcohols conversion

Yangyang Zhang[1,2], Yanxu Chen[1,2], Xiaowen Wang[1], Yafei Feng[1], Zechuan Dai[1], Mingyu Cheng[1] & Genqiang Zhang [1] ✉

The carbon–carbon coupling at the Cu/Cu$_2$O Schottky interface has been widely recognized as a promising approach for electrocatalytic CO$_2$ conversion into value-added alcohols. However, the limited selectivity of C$_{2+}$ alcohols persists due to the insufficient control over rectifying interface characteristics required for precise bonding of oxyhydrocarbons. Herein, we present an investigation into the manipulation of the coordination environment of Cu sites through an in-situ electrochemical reconstruction strategy, which indicates that the construction of low-coordinated Cu sites at the Cu/Cu$_2$O interface facilitates the enhanced rectifying interfaces, and induces asymmetric electronic perturbation and faster electron exchange, thereby boosting C-C coupling and bonding oxyhydrocarbons towards the nucleophilic reaction process of *H$_2$CCO-CO. Impressively, the low-coordinated Cu sites at the Cu/Cu$_2$O interface exhibit superior faradic efficiency of 64.15 ± 1.92% and energy efficiency of ~39.32% for C$_{2+}$ alcohols production, while maintaining stability for over 50 h (faradic efficiency >50%, total current density = 200 mA cm$^{-2}$) in a flow-cell electrolyzer. Theoretical calculations, operando synchrotron radiation Fourier transform infrared spectroscopy, and Raman experiments decipher that the low-coordinated Cu sites at the Cu/Cu$_2$O interface can enhance the coverage of *CO and adsorption of *CH$_2$CO and CH$_2$CHO, facilitating the formation of C$_{2+}$ alcohols.

CO$_2$ electrochemical reduction (CER) to produce value-added chemicals and fuel is an available strategy in response to the growing energy and environmental crisis[1]. C$_{2+}$ alcohols are coveted outputs of CER owing to their extensive market potentials and remarkable energy densities[2]. Indubitably, the harmonious cooperation of biphasic Cu/Cu$_2$O catalyst stands as an eminent contender in engendering C$_{2+}$ alcohols owing to its heightened predilection towards *CO adsorbates

on Cu$^+$ and reduced energy barrier for C–C or C$_2$–C coupling at the rectifying interface[3,4]. Nevertheless, owing to the precarious stability of oxyhydrocarbons intermediates (wherein, C$_2$H$_3$O* serves as the watershed of C$_2$H$_4$ or alcohols) and the oxidation state of Cu, the enduringly biphasic Cu/Cu$_2$O catalyst continues to face significant obstacles in suppressing the desorption of C$_2$H$_3$O* and enhancing the yield of alcohols compared to hydrocarbons in the highly reductive

[1]Hefei National Research Center for Physical Sciences at the Microscale, CAS Key Laboratory of Materials for Energy Conversion, Department of Materials Science and Engineering, University of Science and Technology of China, Hefei, Anhui, China. [2]These authors contributed equally: Yangyang Zhang, Yanxu Chen. ✉e-mail: gqzhangmse@ustc.edu.cn

environmental[5]. Therefore, the feasible strategies are demanded to develop the architectural blueprint of catalysts and electrolytic systems for preserving the oxidation state and bolstering the $CO_2$ performance of copper-based catalysts, including but not limited to elemental doping[6], interface engineering[7,8], intermediate confinement[9], and pulse $CO_2$ electrolysis (P-e$CO_2$R owing the straightforward and readily adjustable means of manipulating anodic potentials for facilitating the formation of Cu oxide species)[10,11]. Especially, the Mott−Schottky catalyst possesses the remarkable ability to hinder the accumulation of electrons, thereby safeguarding the integrity of Cu−O bonds, while simultaneously enabling swift electron transfer courtesy of its built-in electric field[12,13]. Therefore, it necessitates the employment of intricate catalyst configuration and fabrication techniques to elevate the rectifying interface effects of Cu/$Cu_2O$ for the purpose of bonding oxyhydrocarbons.

Several strategies can improve the selectivity of oxyhydrocarbons in CER, including the high concentration of local *CO around the active sites[14,15], doping modification of copper catalysts with heteroatoms[16,17], building of crystal defects and low coordination of copper[12,18,19]. Effectively, the coverage of *CO can be facilitated on the low-coordinated Cu sites of oxide-derived Cu, leading to its subsequent hydrogenation into *COH, which is essential for the coupling of OC−COH[20,21]. For example, Liang group reported a fragmented Cu catalyst with abundant low-coordinated sites by electrochemical reconstruction of B-doped $Cu_2O$, which exhibited a $C_{2+}$ products faradaic efficiency of 77.8% at 300 mA cm$^{-2}$ (see ref. 19). Theoretical computations have demonstrated that the *CO bindings could be strengthened on low-coordinated copper sites and prefer to coupling with *COH instead of *CO dimerization[22]. Moreover, the incorporation of halide species and the creation of oxygen vacancy also contribute to the formation of low-coordinated metal and the controlled generation of intermediates[23,24]. Sun and co-worker intervened the behavior of low-coordination chloride ion (Cl$^-$) adsorption on the surface of a silver hollow fiber (Ag HF) electrode in 3 M KCl electrolyte, which resulted in the high concentration of *CO on low-coordination Ag−Cl state for CER[25]. However, only few studies provide insight on the priority of enhanced rectifying interface effects for bonding oxyhydrocarbons on the low-coordinated Cu/$Cu_2O$ in comparison to the pure Cu/$Cu_2O$

Mott−Schottky catalyst. Thus, the mechanism exploration of low-coordinated Cu/$Cu_2O$ on stabilizing *CO and oxyhydrocarbons intermediates is significant.

Herein, we demonstrated a low-coordinated Cu/$Cu_2O$ Mott−Schottky catalyst with an enhanced rectifying interface (Cu$_L$/$Cu_2O$). This catalyst adjusts the electron densities through interfacial charge exchange, leveraging the difference in Cu and $Cu_2O$ work functions, which can effectively form bonds between *CO and oxyhydrocarbons under CER conditions. Furthermore, Cu$_L$/$Cu_2O$ catalyst generated highly selective catalytic sites for the coupling reaction of *CO−COH and hydrogenation of $C_2H_2O$* intermediate to $C_{2+}$ alcohols. Chlorine-doped cuprous oxide (Cl-$Cu_2O$) and pure $Cu_2O$ were selected as the precursors and electrochemically reconstructed to unsaturated-coordinate Cu$_L$/$Cu_2O$ and pure Cu$_P$/$Cu_2O$. The low-coordinated Cu$_L$/$Cu_2O$ achieved a $C_{2+}$ alcohols faradic efficiency (FE$_{alcohols}$) of 64.15 ± 1.92% with the corresponding energy efficiency of ~39.32%. In addition, a stable $C_{2+}$ alcohols faradaic efficiency of >50% was also obtained during a continuous 50 h chronopotentiometry experiment. Our research provides a platform for the rational design of low-coordinated Cu−$Cu_2O$ Mott−Schottky catalysts and analyzes the key elements for efficient synthesis of $C_{2+}$ alcohols.

## Results
### Synthesis and characterization of Cu$_L$/$Cu_2O$ nanoparticles
The Cu$_L$/$Cu_2O$ catalysts were synthesized using a simple electrochemical reconstruction strategy on Cl-$Cu_2O$ nanoparticles (see details in "Methods", Supplementary Figs. 1–4). The Cu$_L$/$Cu_2O$ catalysts are designed to function as a Mott−Schottky catalyst of Cu/$Cu_2O$, which generates enhanced rectifying interfaces between electron-deficient metal and electron-rich region because of the difference in work function (Fig. 1A). The rectifying interface effect of Cu−$Cu_2O$ Mott−Schottky catalyst adjusts the electronic densities through interfacial charge exchange, resulting in reduced adsorption resistance for intermediates This effect also leads to higher catalytic performance due to the electronic perturbation at both sides of the interface (Fig. 1B)[26]. Enhancement of the rectifying interface effect, however, can be achieved through the smaller size and defects of crystals[27,28], which contributes to faster electron transfer and a denser

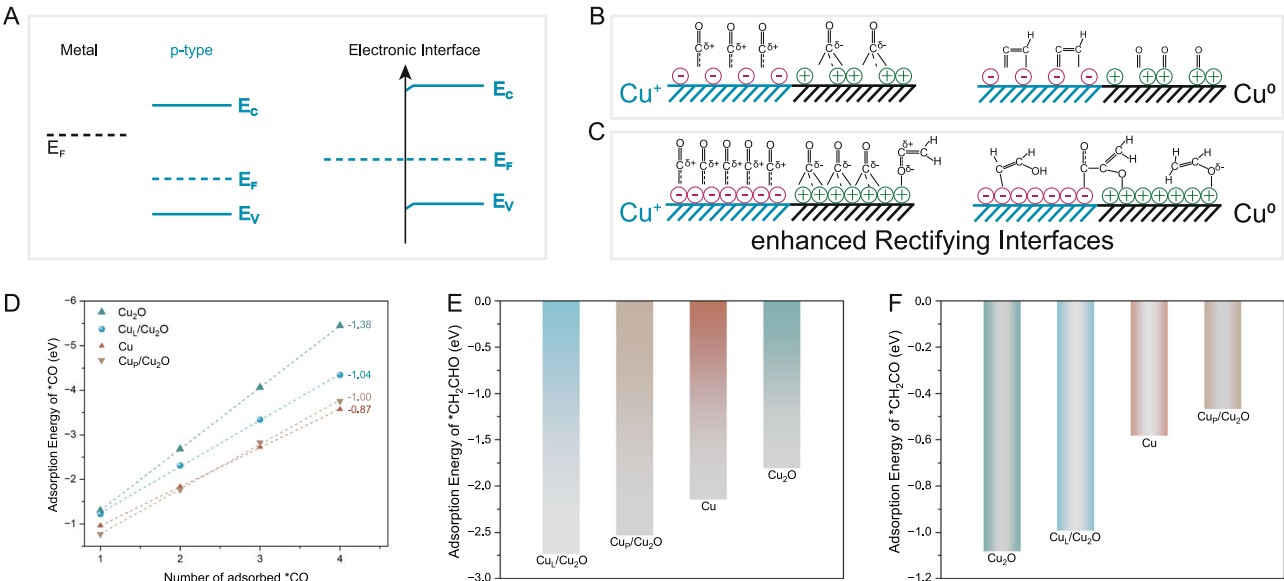

**Fig. 1 | Catalytic mechanism of CuL/Cu2O and free energy of intermediates.**
**A** Schematic diagram of rectifying interfaces in Mott−Schottky catalysts, where $E_F$, $E_C$, $E_V$ are Fermi levels, conduction band and valence band of semiconductors, respectively. **B** The adsorption of intermediates and C−C coupling reaction on Cu$_P$/$Cu_2O$. **C** The adsorption of intermediates and C−C or $C_2$−C coupling reaction on Cu$_L$/$Cu_2O$. **D** Free energy versus the number of adsorbed *CO intermediates on the catalyst models. **E** The formation energy of *$CH_2CHO$ on Cu, $Cu_2O$, Cu$_P$/$Cu_2O$, and Cu$_L$/$Cu_2O$. **F** The formation energy of *$CH_2CO$ on Cu, $Cu_2O$, Cu$_P$/$Cu_2O$, and Cu$_L$/$Cu_2O$.

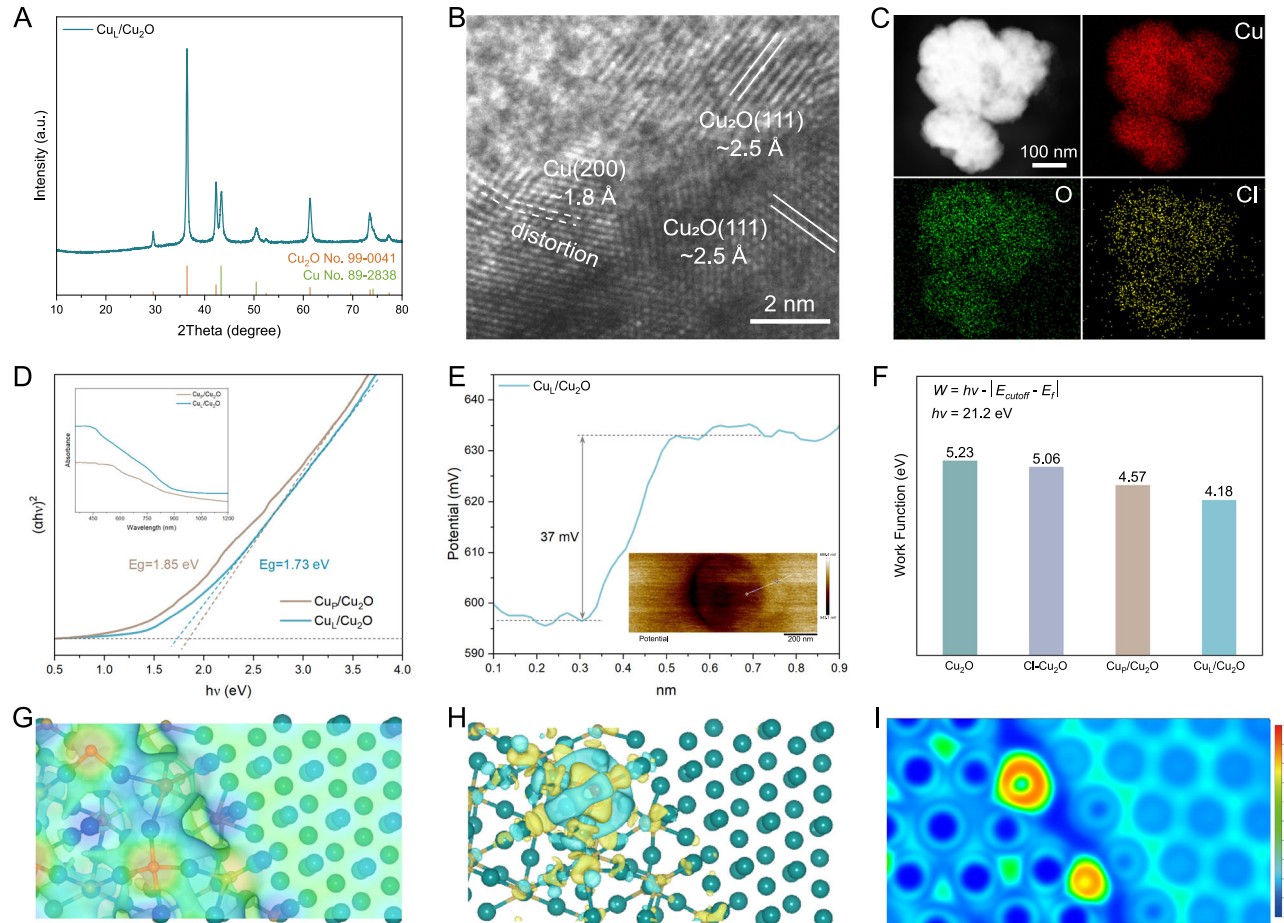

**Fig. 2 | Structures of $Cu_L/Cu_2O$. A** XRD pattern, **B** HRTEM image, and **C** corresponding elemental mapping of $Cu_L/Cu_2O$ showing the interface between Cu and $Cu_2O$ domains. **D** The Tauc plots $(\alpha h\nu)^2$ versus light energy $(h\nu)$ derived by transforming the Kubelka−Munk function on the basis of the inserted UV−vis diffuse absorption spectrum, where Eg is energy bandgap. **E** The surface potential of $Cu_L/Cu_2O$ and corresponding variation of surface potential along with the orientation of the white line, where the gray arrows are the difference between the vertical coordinates corresponding to the two dashed lines and the darker color means lower surface potential. **F** Work function determined by UPS measurements. **G** Electrostatic potential, **H** 3D electron density difference distributions with yellow indicating charge density accumulation and blue indicating deplete and **I** 2D display of electron localization function of $Cu_L/Cu_2O$: red displays high electron density and blue is low.

asymmetric charge distribution (Fig. 1C). Furthermore, the accumulable electron density indicates that the $d \rightarrow 2\pi^*$ back donation induce the transfer of electron density from Cu to the *CO intermediates[29,30]. The adsorption abilities for intermediates on different catalyst models are quantified by DFT calculation (Supplementary Figs. 5−8 and Supplementary Table 3). Figure 1D demonstrates that all these models exhibit a linear relationship between the adsorption energy and *CO coverage. Notably, the $Cu_L/Cu_2O$ model shows a higher stark tuning slope (−1.04), indicating an enhanced interaction between *CO and the catalyst surface, resulting in increased adsorption strength and coverage. In addition, the $Cu_2O$ model bonds more firmly with CO molecules than other models due to the presence of stepped sites, confirming the affinity for *CO adsorbates on $Cu^+$ sites, as stated in previous literature[31,32]. Subsequently, the adsorption energy of $CH_2CHO$ (a branching intermediate for ethanol or ethylene) and $CH_2CO$ intermediates were calculated to assess the potential capacities for ethanol and $C_3$ products (Fig. 1E, F). Both of them exhibit strong adsorption on the surface of $Cu_L/Cu_2O$ model. The *$CH_2CO$ molecular, with the C(2) end of $H_2C(1)C(2)O$ possessing a high positive charge, serves as a key intermediate for the $C_2-C$ coupling reaction, suggesting the $C_2-C$ coupling as a nucleophilic addition reaction process in the mixed-valence boundary region[33]. Thus, the low-coordinated $Cu_L/Cu_2O$ Mott−Schottky catalyst has an enhanced affinity for intermediates compared to pure $Cu_P/Cu_2O$.

To verify the enhanced rectifying interfaces of $Cu_L/Cu_2O$, a series of experiments were conducted. The sharp peaks of XRD patterns matched perfectly with the standard cubic $Cu_2O$ (JCPDS No. 99-0041) and Cu (JCPDS No. 89-2838), providing evidence for the coexistence of $Cu^+$ and $Cu^0$ (Fig. 2A). The SEM and TEM images show that the reconstructed $Cu_L/Cu_2O$ nanoparticles exhibit smaller particle sizes ranging from 200 to 300 nm compared to $Cu_P/Cu_2O$, with no serious agglomeration (Supplementary Figs. 9 and 10). From the high-resolution TEM and SAED images of $Cu_L/Cu_2O$ (Fig. 2B and Supplementary Fig. 10C), the lattice fringes with interplanar spacing of 0.25 and 0.18 nm are classified to the d spacing of exposed $Cu_2O$ (111) and Cu (200) planes, respectively. The distortion of Cu(200) may be caused by the potential surface defect owing to the counter diffusion of lattice O from the $Cu_2O$/Cu interface with low energy barriers[34]. The distorted Cu(200) can induce a low-coordinated Cu and reduce the barriers of intermediates adsorption[35]. Energy dispersive X-ray spectra (EDS) mapping analysis shows the presence of residual chlorine with a content of 1.14 at% and a homogeneous distribution among Cu, Cl, and O (Fig. 2C, Supplementary Fig. 10D and Supplementary Table S2). The enhanced rectifying interfaces of low-coordinated $Cu_L/Cu_2O$ result in an increased absorption edge in the ultraviolet−visible (UV−vis) diffuse reflectance spectrum across the entire wavelength range (inserted in Fig. 2D). Based on the Kubelka−Munk function, the optical bandgap of a semiconductor can be extrapolated from the Tauc plot (the curve of

converted $(\alpha h v)^{1/n}$ versus $hv$ from the UV-vis spectrum, where $\alpha$, $h$, and $v$ are the absorption coefficient, Planck constant, and frequency of the photon, respectively, and $n$ is 1/2 for a direct bandgap semiconductor or 2 for an indirect bandgap semiconductor)[33]. The Tauc plot exhibits a well-matched linear fit with $n = 1/2$, which is in line with previous articles supporting $Cu_2O$ as a direct bandgap semiconductor (Supplementary Fig. 11)[36]. The $E_g$ value of $Cu_L/Cu_2O$ is calculated to be 1.73 eV by measuring the x-axis intercept of an extrapolated line from the linear regime of the curve, which is smaller than that of $Cu_P/Cu_2O$ (1.85 eV). The $Cu_L/Cu_2O$ sample with a minimum band energy and highest conductivity is beneficial for enhancing the rectifying interface effect. Kelvin probe force microscopy (KPFM) was carried out to explore the internal electric field and work function of $Cu_L/Cu_2O$ (Fig. 2E). The contact potential difference (CPD) profile shows light and dark variations due to the different work functions of sample and substrate[37]. According to the CPD profiles, the work function of $Cu_L/Cu_2O$ was defined to be the lowest among the other samples (Supplementary Fig. 12). Figure 2E exhibits the contact potential difference (CPD) profiles for $Cu_L/Cu_2O$, from which an estimate of the average surface potential of 0.037 V can be derived. These results are in accordance with the experimental expectations of Ultraviolet photoelectron spectroscopy (UPS). Importantly, UPS experiments were performed to calculate the work function (WF) of samples for estimating the electron binding factor (Fig. 2F). The experimental WF values for $Cu_2O$, $Cl$-$Cu_2O$, $Cu_P/Cu_2O$, and $Cu_L/Cu_2O$, calculated by recording the secondary electron cutoff and Fermi edge (Supplementary Fig. 13), are 5.23, 5.06, 4.57, and 4.18, respectively. These values suggest that $Cu_L/Cu_2O$ has a smaller escaping resistance for electrons from the surface, which potentially facilitates a synergistic rectifying interface to enhance the asymmetric adsorption of intermediates. The theoretical electrostatic potential energies of the samples are plotted in Fig. 2G, where the surface potentials of the models are calibrated based on the corresponding Fermi levels ($E_{Fermi}$). Noteworthy, the charge transfer at the interface of $Cu_L/Cu_2O$ results in the formation of interface dipole, thereby decreasing the WF of $Cu_L/Cu_2O$ in the vacuum region. The distribution of electron density differences of the internal electric field inside $Cu_L/Cu_2O$ were explored to estimate the potential adsorption of electrically charged intermediates. As shown in Fig. 2H, the electron clouds around the side of $Cu^+$ at the rectifying interface of $Cu_L/Cu_2O$ are much more enriched due to the stronger electronegativity of O than Cl and a low-coordination number than that of $Cu_P/Cu_2O$ (Supplementary Fig. 14 and Supplementary Table 4), indicating the preferred nucleophilic addition reaction process on $Cu_L/Cu_2O$. Due to the difference in electron density, the $Cu^+$ region becomes nucleophilic, while the $Cu^0$ region prefers an electrophilic addition reaction process (Fig. 2I). In conclusion, the faster electron exchange at enhanced rectifying interface regions, resulting from the asymmetric electron aggregation, will boost the nucleophilic or electrophilic addition reaction process for C–C or $C_2$–C coupling. Generally, the enhanced adsorption of negatively charged *CO may focus on the electrophilic Cu sites, while the $C_2$ end of *$C_2H_2O$, with a high positive charge, prefers bonding with nucleophilic $Cu_2O$.

To analyze the origins of enhanced rectifying interfaces, electro paramagnetic resonance (EPR), XPS, and X-ray absorption spectroscopy were utilized to characterize the refined structure of $Cu_L/Cu_2O$. As shown in Fig. 3A, the O $1s$ XPS spectrum is deconvoluted into four peaks situated at 529.7, 530.3, 530.8, and 531.7 eV, corresponding to the lattice oxygen (Cu–O), oxygen vacancies (Vo), C–O bond and adsorbed oxygen on $Cu_L/Cu_2O$ samples, respectively[38]. The ratio of Vo/Cu–O demonstrates a descendant trend with $Ar^+$ etching, implying surface and subsurface oxygen deficiency in the oxide-derived Cu. The shift of bonding energy of Cu–O toward a higher level also indicates the disappearance of Vo within the depth of $Cu_L/Cu_2O$. A characteristic sign of Vo is observed at $g = 2.004$ (Fig. 3B)[39], which supports the

electron-rich status of oxygen-deficient $Cu^+$. However, the $Cu_P/Cu_2O$ exhibits a perfect Mott–Schottky structure without any signs of deficient oxygen (Fig. 3B and Supplementary Fig. 15). From the Cl $2p$ spectrums in Fig. 3C, the bonding energy of Cu–Cl also shifts to a higher state with deeper $Ar^+$ etching due to the presence of sufficient Cl and O atoms in the depth of $Cu_L/Cu_2O$. The clean rectifying interface regions of $Cu_L/Cu_2O$ ensure smooth absorption and desorption, as well as the coupling behavior of oxyhydrocarbon intermediates. In contrast with Cu $2p$ spectra of the $Cu_P/Cu_2O$ species, the Cu $2p$ 3/2 peak of $Cu_L/Cu_2O$ shows a strong characteristic signal for $Cu^0/Cu^+$ at 932.3 eV, accompanied by weak satellite peaks (Fig. 3D). The AES of $Cu_P/Cu_2O$ and $Cu_L/Cu_2O$ were analyzed by Cu LMM Auger spectra to prove the clearer evidence of the valence states of Cu (Fig. 3E). The kinetic energy of the Auger electron transitions of $Cu_L/Cu_2O$ corresponding to $Cu^+$, $Cu^{2+}$, and $Cu^0$ are measured at 916.4, 917.2, and 918.3 eV, respectively, further confirming the coexistence of $Cu^+$ and $Cu^0$ (Supplementary Fig. 16)[40]. Synchrotron-based X-ray absorption fine spectroscopy was employed to analyze the coordination environment of Cu. The adsorption edge position of $Cu_L/Cu_2O$ (8980 eV) is closer to that of $Cu_2O$ (8983.5 eV) and Cu foil, while it is further away from that of CuO, indicating the presence of integral valence $Cu^{\delta+}$ ($0 < \delta < 1$) in the $Cu_L/Cu_2O$ catalyst (Fig. 3F and Supplementary Fig. 17)[41]. Figure 3G shows the Fourier transform (FT) extended X-ray absorption fine structure spectra of $Cu_L/Cu_2O$. A strong signal of the Cu–O bond at 1.5 Å can be detected in $Cu_L/Cu_2O$, accompanied by the Cu–Cu bond at 2.32 Å, which also proves the coexistence of $Cu^0$ and oxidized $Cu^+$. Furthermore, the Cu–O bong length of $Cu_L/Cu_2O$ is slightly longer than that of Cu–O in pure $Cu_2O$ (1.44 Å), implying the presence of Cu–Cl because of the longer Cu–Cl bond compared to Cu–O[42]. Unfortunately, the Cu–Cl sign has fallen out due to the low content of chlorine elements and the shading impact of the strong Cu–O sign. In addition, the $Cu^+$ in $Cu_L/Cu_2O$ may have a low-coordination number due to the deficiency of oxygen and chlorine. To better quantify the coordination number of Cu, EXAFS fitting was carried out (Fig. 3H and Supplementary Table 4). The experimental $k^3$–weight Cu K-edge EXAFS fits perfectly with the calculated R-space fitting model of $Cu_L/Cu_2O$. According to the fitting results, the coordination number of Cu atom of $Cu_L/Cu_2O$ (10.3) is smaller than that of $Cu_2O$ (12.3) and Cu (12), which correspond to the fitting of $Cu_2O$ and Cu, respectively (Supplementary Fig. 18). The structure of the Mott–Schottky catalyst was further analyzed by wavelet transform of the EXAFS. As shown in Fig. 3I, both $Cu_2O$ and $Cu_L/Cu_2O$ have the K values centered around 6.3 $Å^{-1}$, while the value from $Cu_L/Cu_2O$ is located at 7.1, closer to 7.9 $Å^{-1}$ observed from the Cu foil. These analyses confirm that the low-coordinated $Cu_L/Cu_2O$ possesses an electron-rich region and abundant defect sites, which receive nucleophilic addition reaction of C–C or $C_2$–C coupling in the mixed-valence boundary region to generate ethanol and n-propanol.

## Performance of CER in the flow cell

The cathodic electrochemical experiments of $CO_2$ reduction reaction were implemented in flow cells using 1 M KOH as electrolyte under ambient conditions (Supplementary Fig. 19). All working potentials are converted into the reversible hydrogen electrode (RHE) scale with $iR$ correction of 85%. The linear sweep voltammetry (LSV) curves in Fig. 4A show that the $Cu_L/Cu_2O$ catalyst provides higher conductivity and activity for CER, with a total current density ($j_{total}$) of −200 at the applied potential of −0.66 V. The splendid kinetic property of $Cu_L/Cu_2O$ reveals a smaller charge-transfer resistance and higher catalytic activity (tafel slope of 232 mV $dec^{-1}$ and electrochemically active surface area of 6.1 mF $cm^{-2}$) for CER (Supplementary Figs. 20 and 21). Remarkably, $Cu_L/Cu_2O$ shows excellent selectivity of $C_{2+}$ alcohols with a maximum faradic efficiency (FE) of $64.15 \pm 1.92\%$ (ethanol of ~56% and n-propanol of ~8%), whereas $Cu_P/Cu_2O$ only reaches a maximum $FE_{ethylene}$ and $FE_{alcohols}$ of ~38.4% and 11.4%, respectively (Fig. 4B, C and

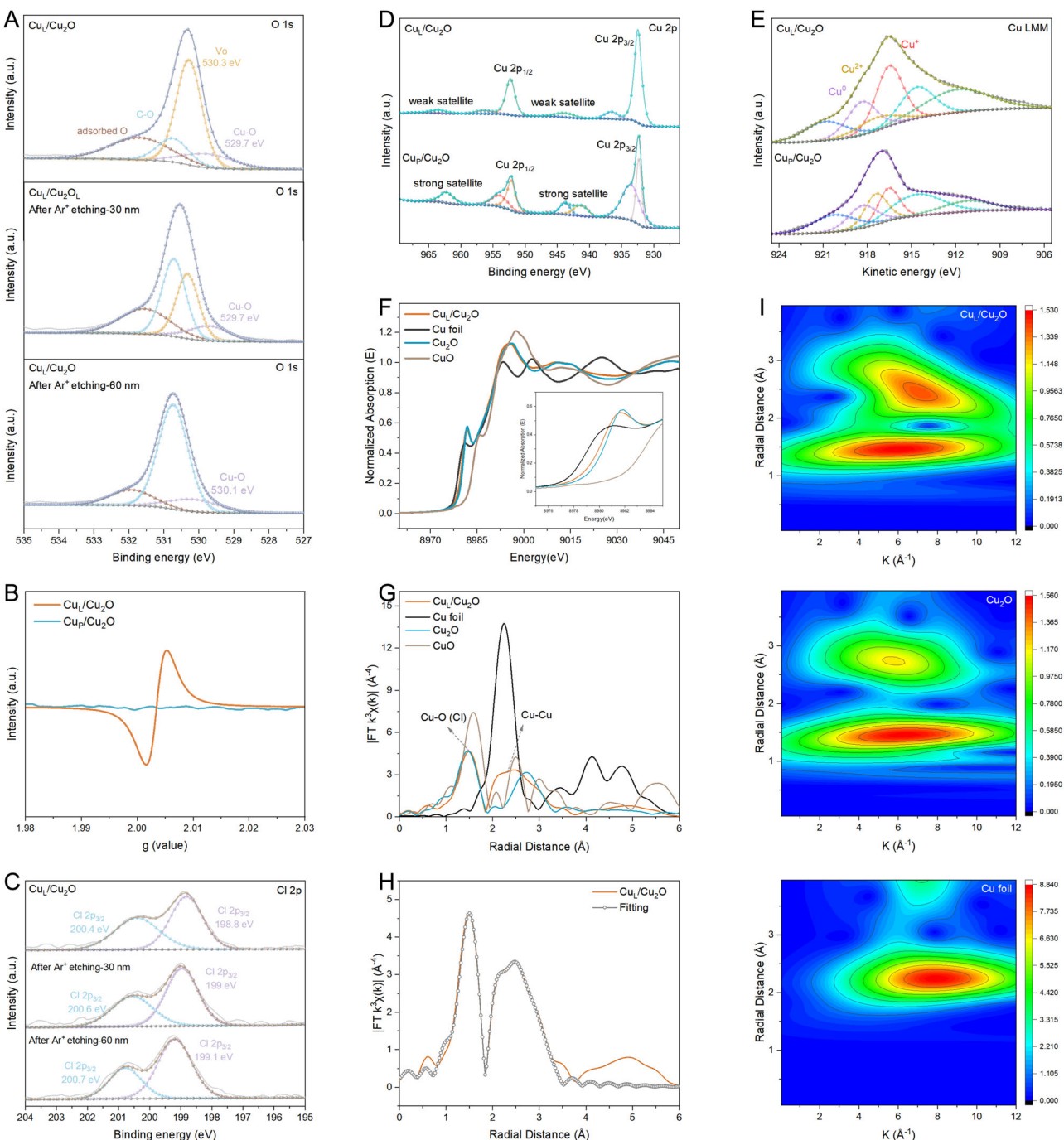

**Fig. 3 | Electron structures and coordination of Cu_L/Cu₂O. A** O *1s* XPS spectra of Cu_L/Cu₂O with or without Ar⁺ etching. **B** EPR spectra of Cu_L/Cu₂O and Cu_P/Cu₂O. **C** Cl *2p*, **D** Cu *2p*, and **E** Cu LMM Auger spectra of Cu_P/Cu₂O and Cu_L/Cu₂O. **F** The normalized Cu K-edge EXAFS spectra and **G** Fourier transform k³-weighted EXAFS for Cu foil, Cu₂O, CuO, and Cu_L/Cu₂O. **H** EXAFS fitting curve of Cu_L/Cu₂O. **I** wavelet transform for the k³-weignted EXAFS for Cu foil, Cu₂O, and Cu_L/Cu₂O.

Supplementary Fig. 22). As shown in Fig. 4D, at a $j_{total}$ of −200 mA cm⁻², the partial current density of alcohols ($j_{alcohols}$) rapidly rises to 128.3 mA cm⁻² on the Cu_L/Cu₂O catalyst (the applied potential is −0.66 V), which is 13-fold higher than that of Cu_P/Cu₂O (9.9 mA cm⁻² at −0.81 V). Meanwhile, the half-cell cathodic energy efficiency (EE) and production rate of alcohols on Cu_L/Cu₂O also reach optimal values of 39.32% and 379.78 μmol cm⁻² h⁻¹, respectively (Fig. 4E, F). Our results feature one of the most excellent efficiencies for alcohol compared with the reported literature to date (Supplementary Table 5). To further investigate the positive effect of halogen anion on CER, comparative experiments with introducing Cl were conducted. The acid-aqueous solution of NH₄Cl resulted in the oxidized Cu²⁺ and physically adsorbed

Cl on the surface of NH₄Cl−Cu₂O (Supplementary Fig. 23). In addition, the distribution of Cl was estimated by XPS after undergoing Ar⁺ etching, confirming the superficial adsorption of Cl (Supplementary Fig. 24 and Supplementary Table 2). The peak values of FE_alcohols and $j_{alcohols}$ on NH₄Cl−Cu₂O are 17.7% and −50.6 mA cm⁻², respectively, indicating a slight increase limited by the negligible Cl concentration. However, a significantly enhanced selectivity of alcohols (a maximum FE_alcohols of 23.5%) is achieved on Cu_P/Cu₂O in 1 M KOH electrolyte with 3 M KCl, suggesting that the specific adsorption of surface-bound Cl, located on the inner Helmholtz planes of Cu electrode, contributes to the C−C or C₂−C coupling owing to the strong chemical affinity of the anion for meal (Supplementary Fig. 25)[43]. Notably, the Cu_L/Cu₂O

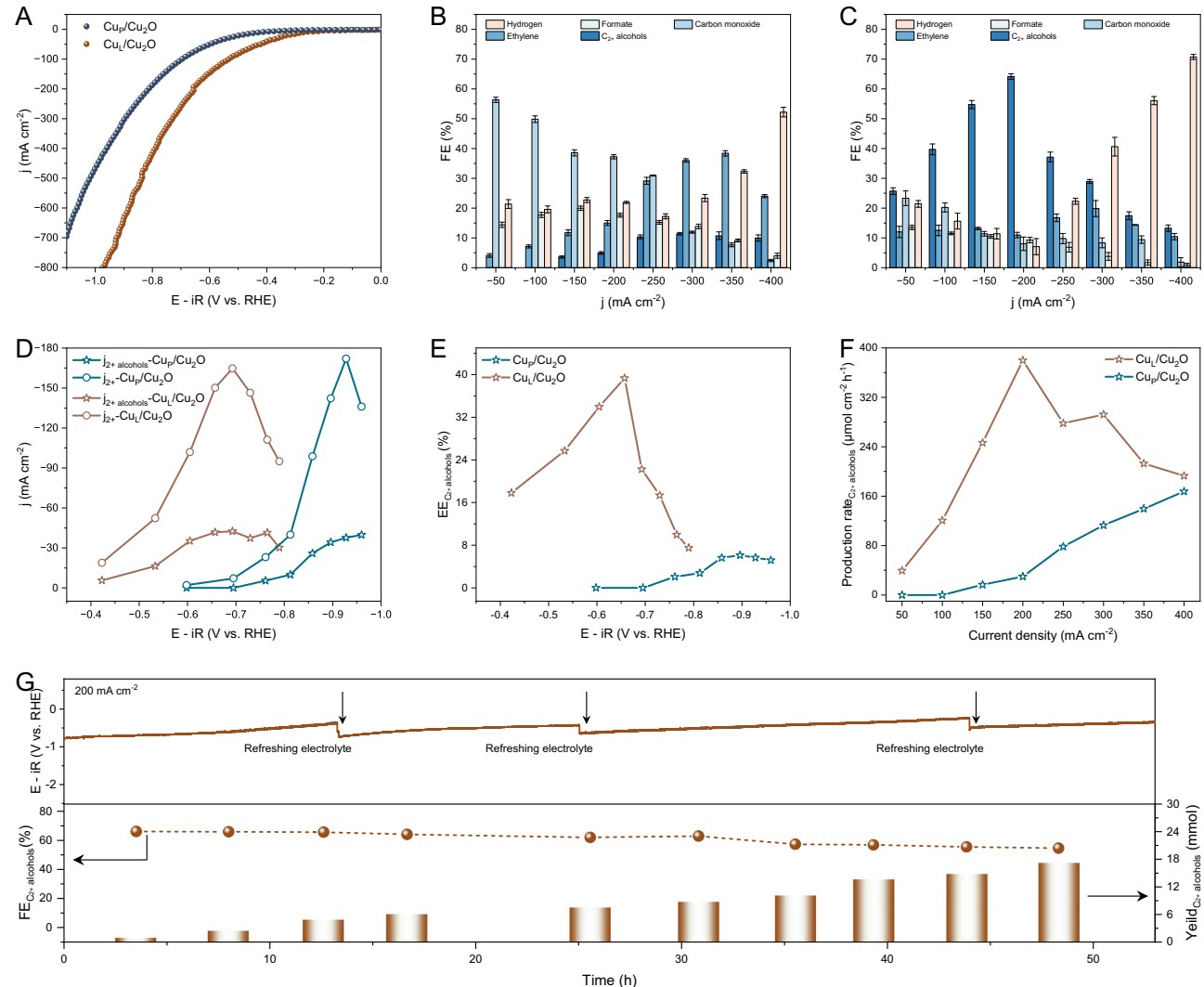

**Fig. 4 | Electrochemical performance of CO₂ reduction. A** LSV curves acquired in flow cell using 1 M KOH as electrolyte, where the curves were calibrated by *iR* compensation with a resistance value of 2.65 ± 0.3 Ω. **B, C** Faradic efficiencies of products at various j$_{total}$ using Cu$_P$/Cu₂O (**B**) and Cu$_L$/Cu₂O (**C**) catalysts. **D** C$_{2+}$ and C$_{2+}$ $_{alcohols}$ partial current density of Cu$_P$/Cu₂O and Cu$_L$/Cu₂O catalysts. **E, F** Energy efficiencies (**E**) and production rate (**F**) of C$_{2+}$ $_{alcohols}$ on Cu$_P$/Cu₂O and Cu$_L$/Cu₂O catalysts. **G** Electrochemical stability test at the j$_{total}$ of 200 mA cm$^{-2}$ using the Cu$_L$/Cu₂O catalysts. Error bars show the standard deviations calculated from three independent experiments.

catalyst still retains its original profile with a mott−schottky interface and unspoiled nanoparticles, as well as an unaltered presence of Cl after CER (Supplementary Figs. 26 and 27). Furthermore, the electrochemical stability of Cu$_L$/Cu₂O for CER was evaluated through chronopotentiometry electrolysis at 200 mA cm$^{-2}$ (Fig. 4G). Despite a steady decline over the course of 50 h electrochemical process, Cu$_L$/Cu₂O achieves a sustained FE$_{alcohols}$ of >50% and an alcohols yield of 17.1 mmol. However, the mention of pulse electrolysis is necessary for long-term industrial production, which may still be an essential strategy for the CO₂ electrolysis community, especially due to its unique advantage in electrolysis stability (>1000 h)[44,45]. The structure of Cu$_L$/Cu₂O maintains a classic architecture of Mott−Schottky catalyst, while slight morphological changes occur (Supplementary Fig. 28). Cl−Cu₂O powders of different particle sizes were prepared and tested under the same electrochemical conditions (Supplementary Figs. 29 and 30). The selectivity of C$_{2+}$ alcohols gradually increases on a gas diffusion electrode loaded with larger-sized particles, which is attributed to the stability of low-coordinated Cu/Cu₂O. The appropriate particle size contributes to the stabilization of the enhanced rectifying interface of the low-coordinated Cu$_L$/Cu₂O Mott−Schottky catalyst. To explore the

effect of electrolytes on the selectivity of C$_{2+}$ alcohols, various electrolytes were used to offer a different pH environment (Supplementary Fig. 31). It undergoes electrochemical reduction reaction of CO₂ with synergistic effect of CO₂ coverage and catalyst in pH-compatible electrolyte and thermodynamics-mediated competitive reaction in a strong alkaline electrolyte. The Cl element shows a positive effect on improving the selectivity of C$_{2+}$ products, but the sensitivity of ethylene is stronger than that of C$_{2+}$ alcohols.

**Structure−property correlation and oxidation state of Cu$_L$/Cu₂O**
Operando FTIR and Raman spectroscopy were employed for investigating the reconstruction of catalysts and adsorbates on both Cu$_L$/Cu₂O and Cu$_P$/Cu₂O. The electrochemical reactions were executed in CO₂-saturated 0.1 M KHCO₃ electrolyte. The reference spectrum was taken at open-circuit potential, and additional spectra were provided in the range of −0.2 ~ −1.1 V vs. RHE. Two bands emerge at 1663 and 1280 cm$^{-1}$ during CO₂ bubbling, which corresponds to the C = O and C−OH stretching vibrations of COOH. These peaks indicate the presence of the COOH* intermediate during the reduction of CO₂ on Cu$_L$/Cu₂O. Furthermore, the C≡O stretch band in the 2000−2100 cm$^{-1}$ range can

be attributable to the linear bonding mode of CO molecules $(CO_L)$[46]. In contrast to the behavior of the $CO_L$ band, the band in the ≈1800–1900 $cm^{-1}$ range, caused by the bridging bonding mode of CO molecules $(CO_B)$, exhibits a significant degree of hysteresis. The red-shift of the $CO_B$ band at more negative potentials is a result of the stark tuning effect[47]. Simultaneously with the CO-related band, a band at 1305 $cm^{-1}$ associated with *COH stretching grows, and *COOH disappears in the spectra. which indicates the performance of $CO_2$ reduction. The absence of the bands at 1589 $cm^{-1}$ in the transmission spectra points out the possibility of the *OCCOH coupling reaction during the reduction of $CO_2$ on $Cu_L/Cu_2O$ (Fig. 5A). However, in contrast to $Cu_L/Cu_2O$ catalyst, some new signals at 1706, 1543, and 1399 $cm^{-1}$ corresponding to the stretch of *CHO, *OCCO, and *OCHO intermediates emerged in spectra of $Cu_P/Cu_2O$, certifying the coupling mode of OC–CO and possible formate product (Supplementary Fig. 32)[48]. Combining with time-independent spectral analysis (Supplementary Fig. 33A, B), apparent peaks are determined at 1064 and 1558 $cm^{-1}$, which correspond well to the symmetric and asymmetric vibrations of the *OCCOH intermediate, respectively[49,50]. In addition, the stretching vibration at 1335 and 1716 $cm^{-1}$ are in close proximity to the stretching of $*OCH_2CH_3$ and *CHO, respectively, which serve as the important intermediates for the subsequent C–C coupling in the production of $C_{2+}$ alcohols[51,52]. These conclusions are consistent with the results obtained from the product analysis of $Cu_P/Cu_2O$ and $Cu_L/Cu_2O$. In the Raman spectra (Fig. 5B and Supplementary Fig. 33D), the $Cu_L/Cu_2O$ powder loaded on carbon paper substrate exhibits typical characteristic peaks of adsorbed *CO intermediates, which are deconvoluted into a low-frequency band (LFB) at ~2045 $cm^{-1}$ and a high-frequency band (HFB) at ~2097 $cm^{-1}$, which verifies the bonding styles on terrace and step sites, respectively[53,54]. A redshift of the $v(CO)$ band indicates the enhanced interaction between the catalyst surface and *CO, resulting in a higher *CO coverage[55]. In addition, the frequencies of $v(CO)$ bands are correlated with the coordination states of Cu sites. Therefore, the low-coordination states of Cu sites in $Cu_L/Cu_2O$ will induce a positive shift of the $d$-band center of Cu and boost the hybridization of the d-band with the $2\pi*$ orbital of CO. The HFB modes derived from the low-coordinated Cu sites will favor the breeding of the $v(C–C)$ band (-1960 $cm^{-1}$), where a blueshift may predict the protonation process C–C bonds[29,30]. On the contrary, the weak signs of the $v(CO)$ and $v(C–C)$ bands on $Cu_P/Cu_2O$ catalyst also verify the poor *CO coverage and C–C coupling (Supplementary Fig. 34B). To verify the origin of the oxidation state of $Cu_L/Cu_2O$, Raman spectroscopy was conducted in the shift range of 200–1600 $cm^{-1}$, and the results are presented in Supplementary Figs. 33A and 34A. In the case of $Cl–Cu_2O$ samples, characteristic signals were observed at 224, 425, 522, and 626 $cm^{-1}$, which were attributed to the $2\Gamma_{12}$, $4\Gamma_{12}$, $\Gamma_{25}$, and $\Gamma_{12} + \Gamma_{25}$ phonon modes, respectively[56,57]. These Raman signals gradually disappear within the potential range from −0.2 to −0.7 V vs RHE, indicating the complete reduction of $Cu_2O$ to metallic Cu. In contrast, the $Cl–Cu_2O$ samples retain the characteristic Raman modes ($2\Gamma_{12}$ and $\Gamma_{12} + \Gamma_{25}$) even at potentials greater than −0.7 V, suggesting that the low-coordinated $Cu_L/Cu_2O$ can protect $Cu^+$ species against reduction due to a combination of environmental and structural factors. Overall, $Cu_L/Cu_2O$ catalyst exhibits a stronger adsorption capacity for *CO due to the restricted rotation and stretching vibration peaks of adsorbed *CO and the Cu–CO bond at around 274 and 358 $cm^{-1}$, respectively[58]. Specifically, the coverage of *CO is closely correlated with the intensity ratio of $v(Cu–CO)$ to $v(*CO)$[59]. Importantly, the intensity ratio of $v(Cu–CO)$ to $v(*CO)$ of $Cu_L/Cu_2O$ is higher in the case of $Cu_L/Cu_2O$ compared to $Cu_P/Cu_2O$, implying a greater CO coverage, which is favorable for the oxidation state of $Cu^+$. The peaks observed at 1015, 1024, and 1069 $cm^{-1}$ can be assigned to the vibrations of $v_1(C–O)$ of $HCO_3^-$, OCO antisymmetric stretching from *COOH, and $v_2(C–O)$ of $CO_3^{2-}$, respectively[60]. Furthermore, as the potential increases, $CO_3^{2-}$ accumulates and disappears on the catalyst surface, indicating the

dissolution and reduction of $CO_2$. Briefly, the origin of the oxidation state of $Cu_L/Cu_2O$ can be summarized as follows: (1) the lattice shrinkage resulting from fast electron transfer from Cu to $Cu^+$ at the enhanced rectifying interface inhibits the loss of oxygen atoms and stabilizes the chemical valence of the cation $Cu^+$ species working with residual Cl atoms[61,62]. (2) The presence of coordination defects and oxygen vacancies increases the adsorption of intermediates rather than forming proton bonds with O atoms of Cu–O to avoid the reduction of $Cu^+$ species[63,64]. (3) Strong electronic interactions occur between carbon intermediates and $Cu^+$ species[65,66]. When CO forms a hybridization orbital with the Cu $d$-states, the $5\sigma$ and $2\pi*$ orbitals of *CO split into bonding and anti-bonding orbitals. In this process, electrons from the Cu d-band are transferred to the $2\pi$ orbitals of *CO through $d$-$2\pi$ back donation, while electrons from the $5\sigma$ orbital of *CO donate to the Cu $d$-band through $5\sigma$-d donation[30]. These effects stabilize the oxidation state of $Cu^+$ species during $CO_2$ electroreduction, allowing for the synergistic effect of Cu and $Cu^+$ species.

The reaction barriers for the hydrogenation process of $C_{2+}$ intermediates on the models of $Cu/Cu_2O$ with and without Cl, respectively, were calculated by DFT. Firstly, by analyzing the Bader charge and bond length of $Cu_L/Cu_2O$ (Fig. 5C, D), it is observed that the lower Bader charge value (0.477 $|e|$) compared to $Cu_P/Cu_2O$ (1.01 $|e|$) indicates that the populated electronic orbitals of low-coordinated Cu sites will easily bind to intermediates and reduce the rate-determining energy barrier[67]. $Cu_L/Cu_2O$ with additionally formed empty orbitals can facilitate electron transfer from hydrogen to nucleophilic intermediates, suggesting the Cu $3d$ orbitals overlap with the $\pi$ orbitals of C and further reduced activation energy for $CO_2$ hydrogenation on $Cu/Cu_2O$[68]. As a result, the nucleophilic addition reaction process of $*CH_2CO$-*CO coupling may effectively react on the mixed-valence boundary region[33], which reduces the formation energy of $*CH_2COCO$ on $Cu_L/Cu_2O$ (−0.148 eV) toward n-propanol (Fig. 5E, Supplementary Fig. 35, and Supplementary Table 6). For the C–C coupling on $Cu_L/Cu_2O$ model (Fig. 5F, Supplementary Fig. 36 and Supplementary Table 7), the *CO–COH coupling is the rate-determining step for $Cu_L/Cu_2O$ catalyst (1.46 eV), indicating the vital rule of *CO coverage in the post-coupled proton-electron transfer (CPET) reaction. The energy barriers of intermediates gradually decrease until $*C_2H_3O$ is reached, which can be further reduced to $*C_2H_4O$ or *O for ethanol or $C_2H_4$, respectively. The $Cu_L/Cu_2O$ model exhibits a more stable adsorption for $C_2H_4O*$ (−0.293 eV) than $Cu_P/Cu_2O$ (−0.154 eV), implying facilitated thermodynamics for increasing the adsorption of key intermediate.

## Discussion

The mentioned theoretical and experimental studies provide tangible proof that the enhanced rectifying interface of low-coordinated $Cu/Cu_2O$ Mott−Schottky catalyst plays an essential role in the selective production of $C_{2+\ alcohols}$. We emphasize several elements of the low-coordinated $Cu_L/Cu_2O$ catalyst for the upgraded CER performance. Firstly, the fast electron exchange and enhanced intermediate adsorption at synergistic rectifying interface regions are due to the low coordination and asymmetric electron aggregation inside the $Cu_L/Cu_2O$ catalyst. In addition, the excellent conductivity and charge difference of $Cu^+/Cu^0$ boost the nucleophilic or electrophilic addition reaction process for C–C or $C_2$–C coupling. Secondly, the fast electron transfer from Cu to $Cu^+$ at the enhanced rectifying interface can induce lattice shrinkage, inhibiting the loss of residual oxygen atoms responsible for stabilizing the chemical valence of $Cu^+$ when working with residual Cl atoms. The clean $Cu/Cu_2O$ regions ensure smooth absorption and desorption, as well as the coupling behavior of the oxyhydrocarbon intermediates. Thirdly, abundant defects originated from oxygen vacancies and residual Cl, as a result of electrochemically reconstructing $Cl–Cu_2O$, supply rich free electrons and landing sites for adsorbed intermediates. These results work together on low-coordinated $Cu_L/Cu_2O$ Mott−Schottky catalyst to improve the selectivity of $C_{2+\ alcohols}$. The faradic efficiency and energy efficiency

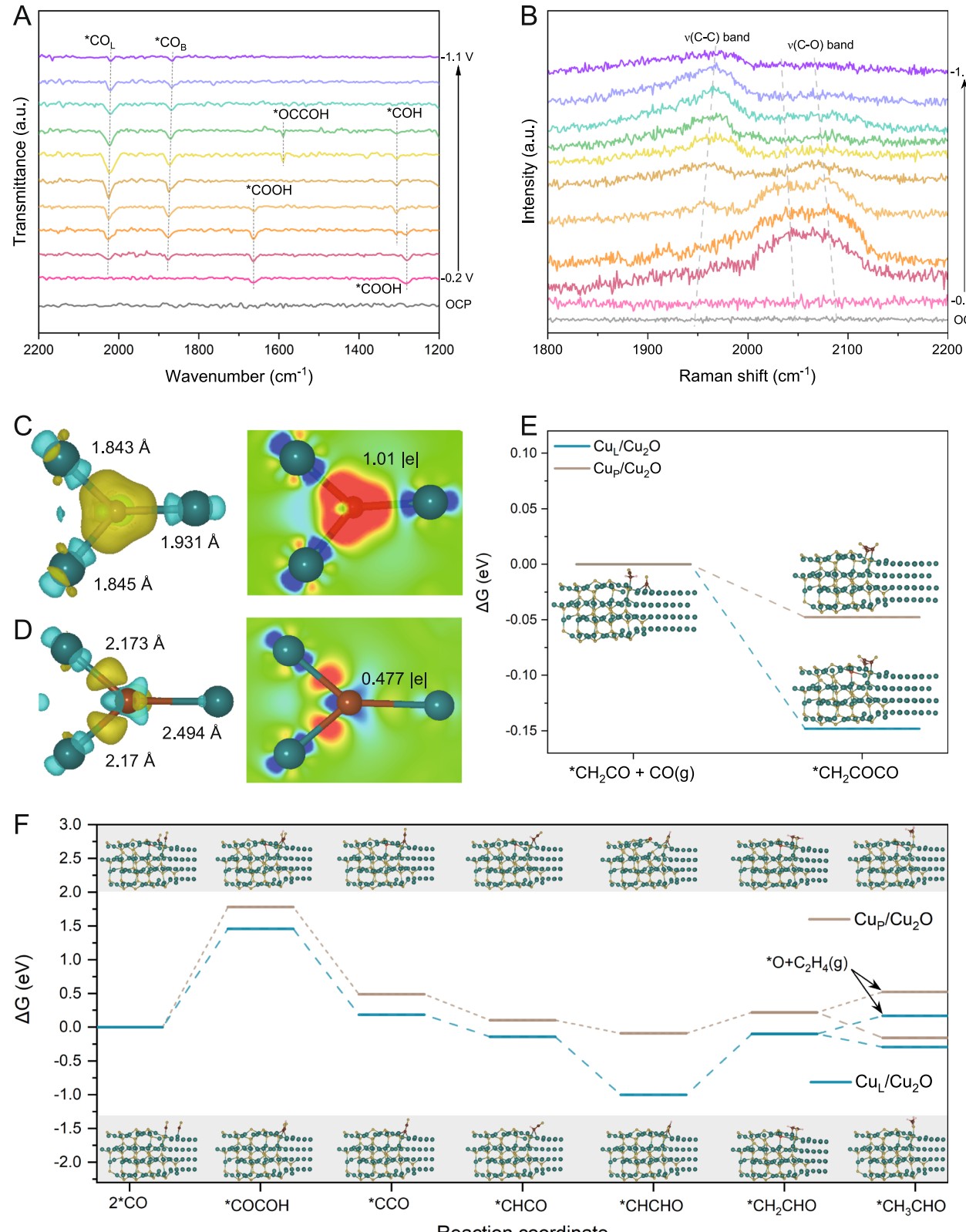

**Fig. 5 | Operando experiments and DFT calculations. A** Operando FTIR and **B** Raman tests of $Cu_L/Cu_2O$ using 0.1 M $KHCO_3$ as electrolyte. **C, D** Bader charge analysis and bond length of $Cu_P/Cu_2O$ and $Cu_L/Cu_2O$, respectively: red and yellow display electron accumulation, blue and cyan indicate electron depletion. **E** Gibbs free energy of $*CH_2COCO$ intermediates for $C_3$ products and **F** C–C coupling to $C_2$ products (ethanol and ethylene) on $Cu_P/Cu_2O$ and $Cu_L/Cu_2O$ catalysts: the steps indicated by the arrows represent the generation of $*O$ and $C_2H_4$.

of $C_{2+}$ alcohols are up to 64.15% and 39.32%, respectively, where the stability of over 50 h ($FE_{C_{2+}\ alcohols}$>50% at $j_{total} = 200\ mA\ cm^{-2}$) explains the positive effects of the above results.

## Methods

### Synthesis of $Cu_P/Cu_2O$, $Cu_L/Cu_2O$, and $NH_4Cl–Cu_2O$ nanoparticles

In the synthesis of the $Cu_2O$ nanocrystals, glucose served as a reductant, contributing to the reduction process[69]. Detail: a mixed solution containing 2 mL of OA and 5 mL of ethanol was added under magnetic stirring for 30 min after dissolving 1 mmol $CuSO_4·5H_2O$ in 15 mL of deionized water and placed in a water bath at 80 °C. Then, 5 mL of 1 M NaOH aqueous solution was added and kept for 10 min. Next, 5 mL of 2 M glucose aqueous solution was added as a reductant to maintain chlorine-free environment for 3 h at 80 °C. The brick red products were obtained by centrifugation and thoroughly cleaned with cyclohexane and ethanol multiple times to eliminate any remaining OA. They were then dried and stored under vacuum at room temperature for future use. As-prepared $Cu_2O$ nanocrystals (2 mg) were mixed with 20 μL of 5 wt% Nafion in a solution composed of ethanol (190 μL) and water (190 μL). The mixture was deposited onto a gas diffusion layer (2 cm × 2 cm) using an airbrush. It was subsequently electrochemically reduced at $-50\ mA\ cm^{-2}$ for 10 min with flowing $CO_2$ in 1 M KOH electrolyte. The obtained catalyst was named as $Cu_P/Cu_2O$, which provided a chlorine-free environment. Using the impregnation approach, a specific quantity of $NH_4Cl$ was loaded onto the $Cu_2O$. In this case, $Cu_2O$ particles were fully immersed in an $NH_4Cl$ solution. To reduce acidity and prevent surface etching, a mixed solution of 80% ethanol and 20% water was used as the solvent. The residuals were then promptly dried and stored for later use in a vacuum at room temperature. After electrochemical reconstruction, the obtained catalyst was named $NH_4Cl–Cu_2O$, which exhibited a surface Cl-modified environment.

In addition, 20 mL of water was used to dissolve 2 mmol of $Cu(CH_3COO)_2·H_2O$. Then, 7 mL of 1 M NaOH aqueous solution was added and kept there while being stirred magnetically for 10 min. After that, 3.5 mL of 2 M $NH_2OH·HCl$ aqueous solution was added as a reductant and kept there for 1 h. The orange products (Cl–$Cu_2O$) were obtained by centrifugation and repeatedly cleaned with methanol and deionized water before being dried and stored for later use under vacuum at room temperature. The Cl–$Cu_2O$ powders with different particle sizes were prepared by modulating the inputs of reactants and reaction time (15 min–5 h). Similar to the electrochemical reconstruction method described above, the only difference is that Cl–$Cu_2O$ nanocrystal as the precursor loaded on carbon paper instead of $Cu_2O$ nanocrystal. The obtained catalyst was named as $Cu_L/Cu_2O$, which showed a chlorine-rich environment.

### Catalyst characterization

X-ray diffraction patterns (XRD) of samples were carried out by TTR-III operating at 40 KV voltage and 15 mA current with Cu Kα radiation ($λ = 0.15406\ nm$). Transmission electron microscopy (TEM), high-resolution TEM, high-angle annular darkfield scanning transmission electron microscopy (HAADF-STEM) and energy disperse spectroscopy (EDS) are recorded on a JEM-2100 (JOEL). Scanning electron microscope (SEM) was performed on FESEM SU8200. Raman data were collected on a Renishaw in Via using 785-nm laser. X-ray photoelectron spectroscopy (XPS) data were collected on Kratos Axis supra + . An $Ar^+$ beam was employed to detect the quantity of Cl and oxygen vacancy with a electron energy of 12.5 V, filament current of 3 mA, emission current of 7.5 A and energy of 2 KV. UPS excitation source: He light source ($hv = 21.22\ eV$), beam spot 2mm. Room temperature UV–Vis absorption was recorded using a Solid 3700 DUV spectrophotometer in the wavelength range of 300–2500 nm. Kelvin

probe force microscopy (KPFM) was performed on Atomic force microscopy (AFM) with a Veeco DI Nanoscope MultiMode V system. Electron Paramagnetic Resonance (EPR) was performed on JES-FA200. Fourier transform infrared (FTIR) spectroscopy was performed with a Thermo-Fisher Nicolet iS10. X-ray absorption near edge structure (XANES) and extended X-ray absorption fine structure (EXAFS) data were collected on beamline 14 W at the Shanghai Synchrotron Radiation Facility. The electro-catalysis actions were tested by CorrTest workstations. The gas chromatographs (GC 7900) equipped with a TCD and FID detector is used to detect the generated gas. Liquid NMR were quantified by the Bruker AVANCE III 400MHZ using dimethyl sulfoxide as an internal standard.

### Operando FTIR and Raman tests

Raman experiments were performed using a Renishaw in Via Raman microscope in a commercial flow cell at the excitation laser source of 785 nm, the electrolyte was 0.1 M $KHCO_3$ aqueous solution. The Cl-doped $Cu_2O$ and $Cu_2O$ precursors were monitored at different potentials and time under the same configuration condition. Then, the intermediates-adsorption measurements were collected signals at different potentials after the precursors were reduced to obtain the $Cu_P/Cu_2O$ and $Cu_L/Cu_2O$. The Raman spectra were collected every 0.1 V at a range of $-0.2 \sim -1.1$ V (vs. RHE) and 150s with bubbling $CO_2$ into the electrolyte. FTIR were measured using a Thermo Scientific Nicolet iS50 FTIR Spectrometer with a Pike VeeMAX III attachment. In total, 2 mg of the catalyst was dispersed in a mixture of 0.38 mL ethanol, and 20 μL 5 wt.% Nafion solution (Sigma-Aldrich) and sonicated, then dropped on carbon paper. Spectra were recorded at different potentials and time in a $CO_2$–saturated 0.1 M $KHCO_3$-$D_2O$ electrolyte. During the Operando FTIR tests, spectra were collected every 0.1 V and 120s with bubbling $CO_2$ into the electrolyte. The spectrum collected at open circuit potential (OCP) in $CO_2$-saturated 0.1 M $KHCO_3$-$D_2O$ electrolyte was used as a background.

### Electrochemical measurements for CER

The experiments were performed in a custom-designed flow-cell system. The carbon paper electrode (the commercial Sigracet 29BC gas diffusion layers with the standard microporous layer based on 77% carbon black and 23% PTFE) was sprayed by airbrush with a loading catalyst of $1\ mg/cm^2$. The geometric area of the electrode was set to $1\ cm^2$. and Ni foam or $IrO_2$/Ti mesh with $IrO_2$ loading of $1\ mg/cm^2$ (for studying the stability of cathode) were acted as working and counter electrode, respectively, which was separated by an anion exchange membrane (Fumasep, FAB-PK-130). The FAB-PK-130 is an anion exchange membrane with a thickness of 130 μm, which is cut to 1.5 cm * 1.5 cm size for practical use. The purchased FAB-PK-130 membrane was placed in 1 M KOH solution for 72 h to activate the anion exchange and subsequently used directly in electrochemical tests. The Hg/HgO electrode was used as a reference electrode. The relevant electrochemical tests were performed in 1 M KOH (or 0.1, 0.5, 1 M $KHCO_3$, 2 M KOH, 3 M KCl) with using a Corrtest Workstation. During the measurements, $CO_2$ was directly fed to the back of cathode GDE at a rate of 20 sccm. The electrolyte was forced to continuously circulate through the chamber at a rate of 10 sccm. All the applied cathode potentials after $iR$ cell compensation were converted to the RHE reference scale using $E_{RHE} = E_{Ag/AgCl} + 0.204V + 0.0591 × pH - 0.85 × i × R$ ($i$: applied current; $R$: cell resistance). The electrochemical impedance spectroscopy (EIS) measurement for measuring the ohmic loss between the working and reference electrodes was performed with frequency ranges from 100,000 to 0.1 Hz and an amplitude of 5 mV at open-circuit voltage in three-electrode system. The linear sweep voltammetry (LSV) curves were carried with a scan rate of $5\ mV\ s^{-1}$. Controlled potential electrolysis was performed at each potential for 10 min.

## CER product analysis

The collected products were analyzed via gas chromatography (GC) and $^1$H NMR on a 400 MHz NMR spectrometer. The gaseous products of $CO_2$ reduction, including carbon monoxide, methane, ethylene, ethane, and propylene, are detected and quantified using GC with an FID detector equipped with a nickel conversion furnace. Hydrogen is quantitatively detected using a TCD detector. The liquid products, including methanol, ethanol, n-propanol, formic acid, and acetic acid, are quantitatively detected using NMR with dimethyl sulfoxide serving as the internal standard. Typically, 0.5 mL KOH electrolyte after electrolysis was mixed with 100 μL of $D_2O$ and 67 μL of DMSO, including 5 mM as an internal standard. The $^1$H NMR spectrum was measured with water suppression via a pre-saturation method. The faradaic efficiency is calculated based on the calibration curve as follows:

$$N_{products} = C_{products} \times V \times N_A \times ne \tag{1}$$

$$N_{total} = \frac{Q}{e} \tag{2}$$

$$FE = \frac{N_{products}}{N_{total}} \times 100\% \tag{3}$$

where $N_{products}$ is total number of product transfer electron, $C_{products}$ is the concentration of product, $V$ is the volume of electrolyte or gases, $N_A$ : avogadro constant, $6.022 \times 10^{23}$ $mol^{-1}$, $n$ is the number of electron transferred for product formation, $e$ is electron, $Q$ is the number of transfer charge, $N_{total}$ is total number of transfer electron.

The energy efficiency (EE) was defined as the ratio of fuel energy to applied energy, which was calculated for the half-cell of CRR with the following equation:

$$EE(\%) = \frac{E^0_{products}}{E^{applied}_{products}} \times FE_{products} \times 100\% \tag{4}$$

Where $E^0_{ethanol}$ = 0.07 V, $E^0_{n-propanol}$ = 0.1 V is the thermodynamic potential of CRR to ethanol and n-propanol, $FE_{products}$ are the faradaic efficiencies of ethanol and n-propanol productions at an applied potential. $E^{applied}_{products}$ is the applied potential for alcohols production.

The production rate for formate was calculated using the following equation:

$$Yield\ rate = \frac{Q \times FE_{alcohols}}{F \times n \times t \times S} \tag{5}$$

where $Q$ is the total charge passed, $t$ is the time (1 h) and $S$ is the geometric area of the electrode (1 cm$^2$).

The partial current densities ($j_{alcohols}$) of products were calculated as below, where $S$ is the geometric area (1 cm$^2$) of the cathode, $i$ is the current of the electrode:

$$j_{alcohols} = \frac{i \times FE_{alcohols}}{S} \tag{6}$$

## Density functional theory (DFT) calculations

All calculations in this study were performed using the Vienna ab initio simulation package (VASP) based on density functional theory (DFT)[70]. We employed projector augmented wave (PAW) pseudopotentials and the Perdew–Burke–Ernzerhof (PBE) exchange-correlation functional within the semi-local generalized gradient approximation (GGA)[71]. To adequately capture weak long-range van der Waals (vdW) interactions, we employed an empirical dispersion-corrected DFT method (DFT-

D3)[72]. The kinetic energy cutoff for the plane wave expansion was set to 500 eV. The self-consistent field (SCF) iteration was considered to converge when the threshold reached $10^{-5}$ eV. Geometry optimization was performed using the conjugate gradient method, with forces on each atom constrained below 0.03 eV Å$^{-1}$.

The $Cu/Cu_2O$ model was obtained by selectively removing specific oxygen atoms from the $Cu_2O(111)$ surface unit cell, adopting a p(4*2) $Cu_2O(111)$ unit cell. Subsequently, we optimized the lattice parameters and atomic positions. The optimized $Cu/Cu_2O$ composite exhibited lattice parameters of $a = 21.48$ Å, $b = 10.29$ Å, $c = 23.62$ Å, $\alpha = 90°$, $\beta = 90°$, $\gamma = 118.63°$. To incorporate chlorine (Cl), an O atom on the surface was replaced with a Cl atom, yielding the $Cl-Cu/Cu_2O$ model. For the $Cu/Cu_2O$ composite slab models, we employed a $1 \times 2 \times 1$ k-point mesh. The $Cu_2O(111)$ model was based on a p(2*2) unit cell, while the Cu(111) model also utilized a p(2*2) unit cell. The atomic coordinates of the optimized computational models are provided in Supplementary Dataset 1, which are defined as Cu-POSCAR, $Cu_2O$-POSCAR, $Cu_P/Cu_2O$-POSCAR, $Cu_L/Cu_2O$-POSCAR. To take into account the on-site Coulomb interaction between 3d electrons of Cu, the GGA +U approach was also employed with a U–J value of 4 eV[73,74]. Note that Cu(111) does not use GGA+U. The reaction free energy change and adsorption energy can be obtained with the equation below:

$$\Delta G = \Delta E + \Delta ZPE - T\Delta S \tag{7}$$

$$\Delta E_{ads} = E_{*X} - E_* - E_X \tag{8}$$

Where $\Delta E$ represents the total energy difference before and after the intermediate is adsorbed, $\Delta ZPE$ and $\Delta S$ denote the differences in zero-point energy and entropy, respectively. The zero-point energy and entropy of the free molecules and adsorbents were derived from vibrational frequency calculations. $E_{*X}$ corresponds to the total energy of the system when molecule $X$ is adsorbed on the surface of the slab, $E_*$ represents the energy of the slab system, and $E_X$ signifies the energy of the adsorbed intermediate $X$.

## Data availability

The raw data of the figures in the main manuscript are available in figshare with the identifier(s) https://doi.org/10.6084/m9.figshare.25124129. All other data needed to evaluate the conclusions in the paper are present in the paper and the Supplementary Information or can be obtained from the corresponding authors upon request. All data are available in the manuscript, the supplementary materials, and from the authors on request. Source data are provided with this paper.

## Code availability

The code used in this work can be obtained from the corresponding authors on request.

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

## Acknowledgements

G.Z. acknowledges the financial support from the National Natural Science Foundation of China (Grant No. 52072359), the Recruitment Program of Global Experts, and the Fundamental Research Funds for the Central Universities (WK2060000016). The numerical calculations in this paper have been done in the Supercomputing Center of the University of Science and Technology of China and TianHe-2 at LvLiang Cloud Computing Center of China. The authors are grateful to infrared beamline (BL01B) at National Synchrotron Radiation Laboratory for the experimental beamtime support. This work was partially carried out at the Instruments Center for Physical Science, University of Science and Technology of China.

## Author contributions

G.Z. conceived and supervised the study. Y.Z. conducted experiments and analyzed data. Y.C. performed and prepared the DFT calculations section. X.W. and Y.F. conducted some experiments. Z.D. and M.C. assisted the electrochemical in situ Fourier Transform Infrared (FTIR) spectroscopy and Raman analysis. G.Z. contributed significantly to the analysis and manuscript preparation. All authors participated in the analysis with constructive discussions.

## Competing interests

The authors declare no competing interests.
