## [Peer Review File · Nature Communications]

REVIEWER COMMENTS

Reviewer #1 (Remarks to the Author):

This manuscript by Zhang et al. presented a novel mechanism that the enhanced rectifying interface of Cu/Cu₂O can boost the generation of C₂+ alcohols, which is inspired for the design and definition of catalyst's structure. It is claimed that the low-coordinated Cu_L/Cu₂O for the building of enhanced rectifying interface can lead to the asymmetric electronic perturbation and faster electron exchange, and the synergistic effect of rectifying interface can effectively increase the C₂+ alcohols selectivity (faradic efficiency of 68.2% and energy efficiency of 41.4%) in flow cell due to the asymmetrical adsorption of oxyhydrocarbons intermediates and boosting C-C coupling on the enhanced rectifying interface. The authors further demonstrated that the enhanced rectifying interface can enhance the coverage of *CO and adsorption of *CH₂CO and CH₂CHO facilitating the formation of C₂+ alcohols based on the experiments and theoretical calculation. Overall, this is an interesting work that providing a promising rationale for CO₂ electroreduction aiming to obtain multi-carbon liquid fuel product. Therefore, the reviewer suggests the acceptance of this manuscript after minor revisions according to the comments listed below:

1. In "Results" section, the manuscript consider that the possible schematic diagram for generating C₂+ alcohols is positive for the enhanced rectifying interfaces shown in Fig. 1B and C. Are there some corresponding studies to prove the conclusions of the schematic diagram? And, in Fig. D, the manuscript shows a linear relationship over the number of adsorbed *CO versus adsorption energy. The physical significance of this fitting slope is important for the linear relationship, the manuscript should clearly indicate it.
2. The second paragraph of "Results", the description of HR-TEM don't give a reasonable explanation about the origin of the distortion of Cu(200), and if the distorted Cu will benefit for the generation of C₂+ alcohols, why it is? In addition, the surface potential based on KPFM can show a lower WF for Cu_L/Cu₂O, the importance of KPFM besides the WF for catalysts?
3. The third paragraph of "Results", the manuscript considers the oxygen vacancy will be existed in Cu_L/Cu₂O, and what induces the generation of oxygen vacancy? why no oxygen vacancy in Cu_P/Cu₂O?
4. the manuscript give a conclusion that the enhanced rectifying interface will boost the generation of C₂+ alcohols. And, the second paragraph of "Results" shows the Cl doping, oxygen vacancy and low-coordinated Cu, so which one leads the enhanced rectifying interface? or works together?
5. The third paragraph of "Results", the description "Briefly, the lattice shrinkage may be resulted by fast electron transfer from Cu to Cu⁺ at the enhanced rectifying interface, which inhibits oxygen atoms loss and stabilizes chemical valence of Cu⁺ working with residual Cl atoms.", why the fast electron transfer will result in a lattice shrinkage? And, why the lattice shrinkage will inhibits oxygen atoms loss?
6. The second paragraph of "Structure-property correlation", the description "the shorter bond length of Cu-O (or Cu-Cl) also implies that electron of d-orbital may be inspired towards to nucleophilic intermediates", what the relationship between the shorter bond length and d-orbital? And, why it will benefit for nucleophilic intermediates?
7. About the interface effect for CO₂ electroreduction to C₂+, the manuscript could refer the description of some studies, such as Adv. Mater. 2022, 34, 2206002; Angew. Chem. Int. Ed. 2022, 61, e202111670; Angew. Chem. Int. Ed. 2021, 60, 10384–10392. And, especially, the article "Perovskite-Socketed Sub-3 nm Copper for Enhanced CO₂ Electroreduction to C₂+" published by Advanced Materials, could provide

some value information for the manuscript.

Reviewer #2 (Remarks to the Author):

This manuscript prepared two different Cu/Cu₂O catalysts by electrochemical reduction of pure Cu₂O and Cl-doped Cu₂O, respectively. Compared with the CuP/Cu₂O catalyst derived from pure Cu₂O, the CuL/Cu₂O derived from Cl-doped Cu₂O exhibited much higher C₂+ alcohol selectivity in electrochemical CO₂ reduction. The superior performance was ascribed to the low-coordinated environment of Cu sites in the as-formed CuL/Cu₂O, facilitating the stabilizing and coupling of some key reaction intermediates. From my point of view, the catalytic performance is at a high level relative to literature results. However, more accurate data and rigorous analysis should be provided to clarify the catalytic structure-property correlation. Detailed comments are listed below:

1. Oxides or other compounds derived Cu catalysts have been extensively studied in recent years, and different CO₂ reduction products have been obtained in these studies. The retention of Cu⁺ or doping elements (e.g., O, Cl) under the harsh CO₂ reduction conditions remains controversial. The present study emphasizes the critical role of Cu⁺ and Cl in CO₂ reduction. The catalysts were formed by a pre-reduction under CO₂ reduction conditions. Why didn't they undergo a further reduction and completely convert into metallic Cu during the CO₂ reduction tests? The pre-formed catalysts have been carefully characterized by many techniques, and the discussion on catalytic mechanism (e.g., DFT calculations) mainly relied on the pristine structures. Figure 2a shows a much higher peak intensity of Cu₂O (111) in comparison to Cu(111). In contrast, the Cu(111) has a higher intensity after the CO₂ reduction test (Figure S25a, S26c, and S27a), suggesting the further reduction and structural reconstruction. Therefore, some in-situ techniques are suggested to identify the real structure or composition in CO₂ reduction, for example, XRD, XAS, Raman.
2. In-situ Raman spectroscopy was employed to characterize the surface reaction intermediate. The spectra at lower Raman shift range (e.g., 200 ~ 1600 cm⁻¹) should be provided and discussed, which may help to probe the catalytic surface reconstruction. For example, the change of copper oxide band at 400~700 cm⁻¹ under different potentials.
3. In page 6 line 3~4, CH₂CHO is claimed as a key intermediate for ethanol. However, it is also regarded as an intermediate of ethylene according to some literature (Adv. Mater. 2021, 2005798).
4. The coexistence of XRD patterns of Cu₂O and Cu cannot prove "the Mott-Schottky interface of metal/metal oxides" (Page 7, line 3-4). The expression is inaccurate because the physical mixing has the same result.
5. The particle sizes of CuL/Cu₂O and CuP/Cu₂O were different. The possible size effect should be excluded to support the discussion on catalytic mechanism.
6. I'm a little confused about the work functions estimated by KPFM (Figure 2E, S12). More discussion on this point is suggested.
7. Cu 2p and LMM spectra should be analyzed through peak deconvolution (Surf. Interface Anal. 2017, 49, 1325), which will help to identify the chemical state of surface Cu. From the shakeup peak in Figure 3E, it seems that the CuP/Cu₂O had a notable amount of Cu²⁺ species, whereas the CuL/Cu₂O did not.
8. Error bars should be added in the CO₂ reduction performance tests.

9. Why would the Cl⁻ ions in the electrolyte adsorb on the negatively-charged electrode surface in CO₂ reduction (Page 14-15)?
10. Some intermediates were detected in the operando FT-IR. However, the catalytic mechanism was not well discussed based on these intermediates. The operando FT-IR of CuP/Cu₂O should also be performed to clarify the catalytic mechanism. Besides, the electrolyte in operando FTIR and Raman tests was different from that in CO₂ performance tests. As the electrolyte was found to have a critical role in CO₂ reduction, the effect of electrolyte should be considered.
11. Some Figures are too blurring to be seen clearly. Besides, there are many clerical errors, for example, Figure 3B legend, CuL/Cu₂O catalyst in Page 16 line 12, sample abbreviation in Figure S2a, standard patterns of Cu and Cu₂O in Figure S27a.

Reviewer #3 (Remarks to the Author):

In this work, Zhang et.al use an electrochemical reconstruction strategy on chlorine doped cuprous oxide to maximize C₂+ alcohols production and report a faradaic efficiency of 68 % at 200 mA/cm². While the catalyst design strategy is quite interesting, the work lacks significant novelty and are inconsistent in explanation at various places which make it unsuitable for publication in Nature Communications journal. This work may be more suitable for a more catalyst focused journal.

1. The authors use Ag/AgCl reference electrode in 1 M KOH electrolyte for CO₂ electrolysis for the prepared Cu catalysts. There have been several reports in the community recommending researchers not to use Ag/AgCl reference electrodes in alkaline media. Hence, all tests must be reported using a Hg/HgO or Hydroflex reference electrode.
2. Carbon based gas diffusion layers (GDL) are used for coating the prepared catalysts. How did the authors mitigate electrolyte flooding at 200 mA/cm² current density? This is especially a common issue with KOH electrolyte and it's surprising that information on this aspect is not provided.
3. Besides, no information on the type of carbon GDL used and PTFE content are provided. In addition, details of the flow cell design are missing.
4. The authors state that Ni foam was used as counter electrode. For the 50 hr test, what was the counter electrode or anode used? The pH of anolyte is well known to drop due to CO₂ crossover, where Ni/NiOOH is known to get corroded due to nickel carbonate formation.
4. The explanation on Matt-Schotky catalyst design is quite confusing and lacks clarity in a few places. How does the chlorine doped cuprous oxide prepared catalyst generate a mix of Cu⁺ and Cu(0) oxidation states during CO₂RR for 50 hrs? This is a big challenge with Cu based catalysts as the oxidation state is well known to reduce to 0 at these negative potentials. Does this mean techniques such as pulsed electrolysis on Cu based catalysts to regenerate Cu⁺ species are not essential compared to this catalyst design ?

5. It is well known that sub-surface oxides are not stable under negative potentials. How does this catalyst (Cu) then generate stable oxide species during CO₂RR? Clarity on what the oxidation state of Cu is during CO₂RR but must be explained.

6. The authors state, “the fast electron transfer from Cu to Cu⁺ at the enhanced rectifying interface can induce lattice shrinkage and inhibit oxygen atoms loss”. In a few lines subsequent to it, it is stated, “Third, abundant defects originated from oxygen vacancies and Cl doping supply rich free electron and landing sites for adsorbed intermediates”. This is quite confusing. Does it inhibit oxygen atoms loss (that is retain oxygen) or create oxygen vacancies?

7. For the preparation of catalyst, the nanoparticles are dispersed in ethanol as the solvent and heated at 80 °C. After this, they also use ethanol to wash it multiple times. Are the authors sure that the ethanol produced from CO₂RR does not come from this probable residual ethanol??

Minor Comments

1. It might be better to report the partial current densities of ethanol instead of the Faradaic Efficiency, especially in Table S5.
2. At some places, the units of current density are missing.

Response to the Reviewers' Comments

Reviewer #1: This manuscript by Zhang et al. presented a novel mechanism that the enhanced rectifying interface of Cu/Cu₂O can boost the generation of C₂₊ alcohols, which is inspired for the design and definition of catalyst's structure. It is claimed that the low-coordinated Cu₁/Cu₂O for the building of enhanced rectifying interface can lead to the asymmetric electronic perturbation and faster electron exchange, and the synergistic effect of rectifying interface can effectively increase the C₂₊ alcohols selectivity (faradic efficiency of 68.2% and energy efficiency of 41.4%) in flow cell due to the asymmetrical adsorption of oxyhydrocarbons intermediates and boosting C-C coupling on the enhanced rectifying interface. The authors further demonstrated that the enhanced rectifying interface can enhance the coverage of *CO and adsorption of *CH₂CO and CH₂CHO facilitating the formation of C₂₊ alcohols based on the experiments and theoretical calculation. Overall, this is an interesting work that providing a promising rationale for CO₂ electroreduction aiming to obtain multi-carbon liquid fuel product. Therefore, the reviewer suggests the acceptance of this manuscript after minor revisions according to the comments listed below:

Author response: We thank the reviewer for the precious time on reviewing our manuscript. We have carefully studied your comments and conducted additional experiments as well as analyses accordingly in order to gain deeper understanding about the underlying mechanism, which further enhance the quality of our manuscript. We believe that the quality of our manuscript has been improved by our provision of more experimental evidence and the details are provided below.

[1] In “Results” section, the manuscript consider that the possible schematic diagram for generating C₂₊ alcohols is positive for the enhanced rectifying interfaces shown in Fig. 1B and C. Are there some corresponding studies to prove the conclusions of the schematic diagram? And, in Fig. D, the manuscript shows a linear relationship over the number of adsorbed *CO versus adsorption energy. The physical significance of this fitting slope is important for the linear relationship, the manuscript should clearly indicate it.

Response: Thank you for your valuable comments. According to the corresponding studies, the schematic diagram presents that the Cu⁺ and Cu⁰ surface sites are responsible for more efficient

adsorption of intermediates because of locally enhanced electric field, which can raise the *CO intermediates and facilitate C-C or C_2-C coupling for high-efficiency conversion of CO_2 to C_2+ alcohols (PNAS, 2017, 114, 26, 6685–6688; J. Phys. Chem. C 2022, 126, 5502–5512). For examples, the C_2 end of H_2CCO possesses a high positive charge.

Figure. Free energy profiles (at $U=-0.9$ V) of CO dimerization in the MM (blue), FOM (red), and MEOM (green) models and for CO hydrogenation to form surface CHO species in the MEOM model (gray green) at pH 7 (PNAS, 2017, 114, 26, 6685–6688).

Figure. The schematic representations of the ELF plots and Bader charge analysis of H_2CCOCO (Left) and $HCCHOCO$ (Right) (J. Phys. Chem. C 2022, 126, 5502–5512).

The Fig. 1D of the manuscript shows a linear relationship over the number of adsorbed *CO versus adsorption energy. The physical significance of this fitting slope is the adsorption strength and coverage of *CO intermediates, which implies the potential advantage of C-C or C_2-C coupling. The relevant revision have been corrected in the manuscript. Please see detailed information marked in red (Results, Paragraph 1).

[2] The second paragraph of “Results”, the description of HR-TEM don’t give a reasonable explanation about the origin of the distortion of Cu(200), and if the distorted Cu will benefit for the generation of C₂₊ alcohols, why it is? In addition, the surface potential based on KPFM can show a lower WF for Cu₁/Cu₂O, the importance of KPFM besides the WF for catalysts?

Response: We thank the reviewer for the professional suggestion, which is very important for us to deeply understand the relationship between catalytic performance and Cu. Based on the previous studies, the distortion of Cu(200) may be induced by the potential surface defect owing to the counter diffusion of lattice O from the Cu₂O/Cu interface with low energy barriers (Nature 2022, 607, 708-713). The distorted Cu(200) can induce a low-coordinated Cu and reduce the barriers of intermediates adsorption (Nat. Commun. 2023, 14, 4882). The surface potential based on KPFM can estimate the direction of built-in electric field and the transport mechanism of photoelectrons, but the WF is a key parameters for electrocatalysts in this manuscript. According to your suggestion, and the modification is marked red in the revised manuscript.

The topography and surface potential map of samples were characterized simultaneous by using KPFM. The schematic diagram of the KPFM system is shown in Figure (J. Mater. Chem. A, 2020, 8, 8586-8592). The topographies and surface potentials were measured by KPFM based on dynamic force microscopy principles to give work functions of the samples. The work function ϕ_{sample} was further calculated from the surface potential according to equation:

$$\phi_{\text{Sample}} = \phi_{\text{Tip}} + qV_{\text{CPD}}$$

$$V_{\text{CPD}} = V_{\text{CPD-sample}} - V_{\text{CPD-substrate}}$$

$$\phi_{\text{Sample}} = \phi_{\text{substrate}} + qV_{\text{CPD}}$$

where $V_{\text{CPD-sample}}$, $V_{\text{CPD-substrate}}$ are the contact potential difference of sample and substrate measured by KPFM in volts. ϕ_{Tip} , ϕ_{sample} and $\phi_{\text{substrate}}$ are the work functions of tip, sample and substrate in eV, respectively and q is the electronic charge. The work functions of tip can be calculated by measuring gold samples by KPFM.

Figure. The schematic diagram of the KPFM system (J. Mater. Chem. A, 2020, 8, 8586-8592).

[3] The third paragraph of “Results”, the manuscript considers the oxygen vacancy will be existed in $\text{Cu}_L/\text{Cu}_2\text{O}$, and what induces the generation of oxygen vacancy? why no oxygen vacancy in $\text{Cu}_P/\text{Cu}_2\text{O}$?

Author response: We really appreciate the reviewer for the professional and meaningful suggestion regarding additional evidence to support our claimed the origin of oxygen vacancy in the manuscript, which is quite helpful and significant for us to further improve the quality of our manuscript. According to the reviewer’s suggestion, we have reviewed the explanation about the generation of oxygen vacancy owing to the deficiency of O atoms around Cu^+ sites. The EPR spectra of Cl- Cu_2O is supplied in Supplementary Figure 1, where the deficiencies of oxygen atoms are derived from the introduce of Cl. As shown in Supplementary Figure 26(A), Fig. 3 (B) and Supplementary Table 2, the $\text{Cu}_L/\text{Cu}_2\text{O}$ catalyst can hold a larger proportion of Cu^+ than $\text{Cu}_P/\text{Cu}_2\text{O}$ owing to the residual Cl and O atoms during the electrochemical reconstruction of Cl- Cu_2O , which creates many oxygen vacancy because of the introduce of Cl and reconstruction of catalyst. On the other hand, the existence of oxygen vacancy ensures the stability of Cu^+ in $\text{Cu}_L/\text{Cu}_2\text{O}$ (J. Am. Chem. Soc. 2023, 10.1021/jacs.3c08312). The absence of signals of oxygen vacancy for $\text{Cu}_P/\text{Cu}_2\text{O}$ is limited by the poor stability of Cu^+ during the electrocatalysis. The related modifications are marked red in the revised manuscript.

Supplementary Figure 26. (A) XRD patterns of $\text{Cu}_\text{P}/\text{Cu}_2\text{O}$ and $\text{Cu}_\text{L}/\text{Cu}_2\text{O}$ catalysts after CER.

Fig. 3 (B) EPR spectra of $\text{Cu}_\text{L}/\text{Cu}_2\text{O}$ and $\text{Cu}_\text{P}/\text{Cu}_2\text{O}$.

Supplementary Table 2. The surface composition of samples based on the conclusion of XPS.

element	O (at%)	Cu (at%)	Cl (at%)
Cu ₂ O	38.94	61.06	-
Cl-Cu ₂ O	24.57	72.14	3.29
Cl-Cu ₂ O After Ar ⁺ etching-30 nm	22.76	73.8	3.43
Cl-Cu ₂ O After Ar ⁺ etching-60 nm	21.3	75.06	3.64
NH ₄ Cl-Cu ₂ O	72.16	19.8	8.04
NH ₄ Cl-Cu ₂ O After Ar ⁺ etching-30 nm	15.84	83.23	0.93
Cu _p /Cu ₂ O	15.2	84.8	-
Cu _l /Cu ₂ O	120.84	78.22	1.14
Cu _l /Cu ₂ O After Ar ⁺ etching-30 nm	21.46	77.05	1.49
Cu _l /Cu ₂ O After Ar ⁺ etching-60 nm	22.14	76.22	1.64

Supplementary Figure 2 | The refined structure of Cl-Cu₂O. (A) Powder XRD profiles and Rietveld refinement patterns and corresponding ball model of Cl-Cu₂O. (B) EPR spectra of Cl-Cu₂O. (C) TEM image, (D) SAED, (E) HRTEM analysis and (F) corresponding elemental mapping results.

[4] The manuscript give a conclusion that the enhanced rectifying interface will boost the generation of C₂₊ alcohols. And, the second paragraph of “Results” shows the Cl doping, oxygen vacancy and low-coordinated Cu, so which one leads the enhanced rectifying interface? or works together?

Author response: We appreciate the reviewer for the professional question. The enhanced rectifying interface can be achieved by the design of Mott-Schottky catalysts. According to the Poisson equation (J. Catal. 1996, 161, 560–569), the difference in the work functions between the metal and semiconductor ($\Phi_M - \Phi_S$) is the main driving force of the interfacial charge redistribution. Additionally, rationally increasing the dielectric constant (ϵ_S) and/or decreasing the dopant concentration (N_d) of an n-type semiconductor could extend the width of the depletion region (W). On one hand, a clean and stable interface is so important. For example, the thin added layer of organic molecules between the metal and semiconductor may completely disturb the rectifying contact and hindrance the effective charge exchange between the metal and semiconductor (Nanoscale 2017, 9, 9440–9446). On other hand, he rational combination of a metal and semiconductor with suitable dielectric constants, work functions and doping concentrations can enhance the rectifying contact, generating more pronounced electron-rich and electron-depletion regions (J. Catal. 1996, 161, 560–569. Chem. Commun. 2019, 55, 11394–11397. J. Am. Chem. Soc. 2022, 144, 5418–5423. Chemistry 2014, 20, 16732–16737). And, the low-coordinated Cu in $\text{Cu}_L/\text{Cu}_2\text{O}$ catalyst is derived the Cl doping and oxygen vacancy. Thus, the low-coordinated $\text{Cu}_L/\text{Cu}_2\text{O}$ catalyst can expose the enhanced rectifying interface to the reaction atmosphere and maximize the contribution of interfacial synergy to the promoted activity of a Mott–Schottky catalyst. The related modifications are marked red in the revised manuscript.

Figure. Schematic structure of a Schottky heterojunction composed of a metal and n-type semiconductor (J. Catal. 1996, 161, 560–569).

[5] The third paragraph of “Results”, the description “Briefly, the lattice shrinkage may be resulted by fast electron transfer from Cu to Cu^+ at the enhanced rectifying interface, which inhibits oxygen

atoms loss and stabilizes chemical valence of Cu^+ working with residual Cl atoms.”, why the fast electron transfer will result in a lattice shrinkage? And, why the lattice shrinkage will inhibits oxygen atoms loss?

Author response: According to the reviewer's suggestion, we carefully investigate the studies about lattice shrinkage. The electronic component of thermal conductivity is determined by the Lorenz factor (L) and electrical conductivity (σ) through the Wiedemann–Franz law ($\kappa_E = L\sigma T$, where T is temperature), and thus improving σ by increasing the solid carrier concentration can effectively enhance the κ_E . Owing to the interaction between electrons and lattice vibration, when the lattice atomic vibration is offset (scattering takes place), there is an exchange of momentum and energy between electrons and lattice vibration, resulting in coupling between electrons and phonons. But for many non-metallic materials with low electrical conductivity, κ_E makes only a trivial contribution to the total κ , and κ_L dominates the total κ . According to the Boltzmann transport theory of phonons, κ_L is determined by the specific heat (C_V), phonon group velocity (v_g), and phonon relaxation time (τ), which are all strongly linked to the dispersion of phonons, including acoustic phonons and optical phonons. Thus, an improvement in electronic conduction can result in a stronger lattice vibration (ACS Energy Lett. 2023, 8, 4540–4546).

To stabilize the oxidation states of metals, the compressive strain can restricts oxygen migration for protecting the oxidation state of the desired Cu^+ sites (Adv. Funct. Mater. 2016, 26, 1564–1570. ACS Nano 2013, 7, 3276–3286). The interphase strain between Cu and oxidized Cu will be enhanced by lattice shrinkage, which will further inhibit oxygen loss (J. Am. Chem. Soc. 2023, 145, 8714–8725). The related modifications are marked red in the revised manuscript.

[6] The second paragraph of “Structure-property correlation”, the description “the shorter bond length of Cu-O (or Cu-Cl) also implies that electron of d-orbital may be inspired towards to nucleophilic intermediates”, what the relationship between the shorter bond length and d-orbital? And, why it will benefit for nucleophilic intermediates?

Author response: Thank you for your valuable comments. The shorter bond length implies a stronger interaction between Cu and O (Cl). The Cu 3d orbitals overlap with the π orbitals of C, indicating that the adsorbate state is mainly ruled by d- π interactions. The d-band center (ϵ_d) of Cu/Cu₂O is narrower and shifted upwards compared with Cu, contributing to the formation of d- π

antibonding orbitals above the Fermi level (ϵ_F). The additionally formed empty orbitals could facilitate electron transfer from hydrogen to nucleophilic intermediates, and suggest a further reduced activation energy for CO₂ hydrogenation on Cu/Cu₂O. The related modifications are marked red in the revised manuscript.

Figure. Projected electronic densities of states of the C 2p orbitals of C₂H₂ in a vacuum, and of the Cu 3d orbitals and the C 2p orbitals of C₂H₂ adsorbed on Cu(200) and Cu/Cu₂O (Nat. Cat. 2021, 4, 565-574).

[7] About the interface effect for CO₂ electroreduction to C₂⁺, the manuscript could refer the description of some studies, such as Adv. Mater. 2022, 34, 2206002; Angew. Chem. Int. Ed. 2022, 61, e202111670; Angew. Chem. Int. Ed. 2021, 60, 10384–10392. And, especially, the article “Perovskite-Socketed Sub-3 nm Copper for Enhanced CO₂ Electroreduction to C₂⁺” published by Advanced Materials, could provide some value information for the manuscript.

Author response: We thank the reviewer for pointing out the shortcomings of manuscript. According to the reviewer's suggestion, we carefully check manuscript and clarify the conclusion about the “Introduction”. In the revised manuscript, we have updated the explanation and corresponding references marked in red.

Reviewer #2: This manuscript prepared two different Cu/Cu₂O catalysts by electrochemical reduction of pure Cu₂O and Cl-doped Cu₂O, respectively. Compared with the Cu_p/Cu₂O catalyst derived from pure Cu₂O, the Cu_L/Cu₂O derived from Cl-doped Cu₂O exhibited much higher C₂+ alcohol selectivity in electrochemical CO₂ reduction. The superior performance was ascribed to the low-coordinated environment of Cu sites in the as-formed Cu_L/Cu₂O, facilitating the stabilizing and coupling of some key reaction intermediates. From my point of view, the catalytic performance is at a high level relative to literature results. However, more accurate data and rigorous analysis should be provided to clarify the catalytic structure-property correlation. Detailed comments are listed below:

Author response: Thanks again for the reviewer's precious time and professional suggestions, which are very important as an insightful guidance for us to further improve the quality of our manuscript. We sincerely hope that our revisions according to the reviewer's suggestions can provide more detailed data and analysis to clarify the catalytic structure-property correlation. We really appreciate your precious time on reviewing our revised manuscript.

[1]. Oxides or other compounds derived Cu catalysts have been extensively studied in recent years, and different CO₂ reduction products have been obtained in these studies. The retention of Cu⁺ or doping elements (e.g., O, Cl) under the harsh CO₂ reduction conditions remains controversial. The present study emphasizes the critical role of Cu⁺ and Cl in CO₂ reduction. The catalysts were formed by a pre-reduction under CO₂ reduction conditions. Why didn't they undergo a further reduction and completely convert into metallic Cu during the CO₂ reduction tests? The pre-formed catalysts have been carefully characterized by many techniques, and the discussion on catalytic mechanism (e.g., DFT calculations) mainly relied on the pristine structures. Figure 2a shows a much higher peak intensity of Cu₂O (111) in comparison to Cu(111). In contrast, the Cu(111) has a higher intensity after the CO₂ reduction test (Figure S25a, S26c, and S27a), suggesting the further reduction and structural reconstruction. Therefore, some in-situ techniques are suggested to identify the real structure or composition in CO₂ reduction, for example, XRD, XAS, Raman.

Author response: We thank the reviewer for the good suggestion. Firstly, the OD-Cu catalysts assigned by some authors to the presence of surface Cu^{δ+} sites can greatly improve faradaic efficiency and selectivity toward C₂ products as compared to polycrystalline Cu (J. Am. Chem.

Soc. 2019, 141, 7646–7659; Nat. Chem. 2019, 11, 222–228; Adv. Mater. 2023, 35, 2208996). Surface analysis and depth profiling of the oxygen concentration in the electrode detected a residual oxide layer in the surface region, which led to the claim that both the oxide-derived metallic layer and the surface oxide are key reaction sites in the selective CO₂ catalysis toward multicarbon products. In certain cases, spectroscopic techniques, such as operando X-ray absorption spectroscopy and scanning transmission electron microscopy showed that Cu⁺ species remain on the surface during the reduction reaction and that they are the active species for reducing CO₂ to ethylene (Nat. Commun. 2016, 7, 1–9). However, the existence of Cu⁺ sites was challenged by Ager and co-workers, which could not detect Cu⁺ under reducing CO₂ conditions (Angew. Chem., Int. Ed. 2018, 57, 551–554; ACS Appl. Mater. Interfaces 2018, 10, 8574–8584; Nat. Catal. 2023, 6, 837–846). The metallic Cu⁰ formed during in situ transformations from the copper oxides and not the copper oxide itself are the active catalyst species in CO₂RR. Merely, ¹⁸O isotope labeling and secondary-ion mass spectrometry measurements showed that the residual oxides are not present in significant amount during the CO₂RR (Angew. Chem., Int. Ed. 2018, 57, 551–554; ACS Appl. Mater. Interfaces 2018, 10, 8574–8584).

The importance and strategies of having both Cu⁰ and Cu⁺ at the surface or subsurface of the Cu catalysts was demonstrated using a different approach. A combined experimental and theoretical study by Sargent and co-workers showed that higher stability and selectivity toward C₂ products can be achieved by the inclusion of boron as a doping element on both Cu(100) and Cu(111) surfaces (Nat. Chem. 2018, 10, 974–980). Ethylene and ethanol are the major hydrocarbons being produced with a remarkable maximum total Faradaic efficiency of ~80% (53% ethylene and 27% ethanol). Because of a partial electron transfer from copper to boron, such Cu atoms adjacent to B atoms are more positively charged. Therefore, the B-doped copper system has both Cu⁺ and Cu⁰ sites, exhibiting a motif analogous to the Cu₂O/Cu catalyst of Goddard et al. discussed in previous study (Proc. Natl. Acad. Sci. U. S. A. 2017, 114, 6685–6688). The ability to tune the average oxidation state of copper enables control over CO adsorption and dimerization. Indeed, the CO adsorption energy increases linearly with the copper oxidation state, and the average copper valence state for the highest Faradaic efficiency was estimated as +0.35. This novel boron-doped catalyst suppresses the C1 reaction pathway by increasing the barrier of the rate-limiting *CO + *H → *CHO step, whereas it enhances C₂ product formation by decreasing the

barrier for the *CO dimerization step. Besides, Han and co-workers have reported an operationally simple in situ dual doping strategy to construct efficient CO₂-to-methanol electrocatalysts. In particular, experimental and theoretical studies suggest that the anion S can effectively adjust the electronic structure and morphology of the catalysts in favor of the methanol pathway, whereas the cation Ag suppresses the hydrogen evolution reaction. Their synergistic interactions with host material enhance the selectivity of methanol and structural stability (Nat. Commun. 2022, 13:1965). In addition, Yan and workers have decorated Cu₂O nanoparticles with hexagonal boron nitride (h-BN) nanosheets to stabilize Cu⁺. Experimental and theoretical studies confirmed strong electronic interactions between the two components in Cu₂O–BN, which strengthens the Cu–O bonds. Electrophilic h-BN receives partial electron density from Cu₂O, protecting the Cu–O bonds from electron attack during the CO₂RR and stabilizing the Cu⁺ species during long-term electrolysis. The well-retained Cu⁺ species enhanced the C₂ product selectivity and improved the stability of Cu₂O–BN (Angew. Chem., Int. Ed. 2022, 61, e202205832).

Figure. Preparation and characterization of Cu(B). (a) Schematic of the wet-chemical process to synthesize Cu(B) samples. (b, c) Boron XPS spectrum and dissolving-time-dependent boron concentrations (Nat. Chem. 2018, 10, 974–980).

Figure. Oxidation state of copper in Cu(B) samples. (a) Copper K-edge XANES spectra of Cu(B) samples after being electrochemically reduced. (b) Average oxidation state of copper in Cu(B). (c) In situ copper K-edge XANES spectra of Cu(B)-2 immediately after CV reduction (Nat. Chem. 2018, 10, 974–980).

Figure. In situ Raman spectra of a) Cu₂O and d) Cu₂O-BN as a function of reaction time at -0.4 V. XRD patterns of b) Cu₂O and e) Cu₂O-BN after different electrolysis time at -1.4 V. Quasi in situ XPS profiles of c) Cu₂O and f) Cu₂O-BN after different electrolysis time at -1.4 V. g, h) HRTEM images of two selected regions of the used Cu₂O-BN after 2 h of electrolysis at -1.4 V. i, j) Elemental mappings of two isolated Cu₂O-BN particles after 2 h of electrolysis at -1.4 V (Angew. Chem., Int. Ed. 2022, 61, e202205832).

In this manuscript, Operando SR-FTIR and Raman measurements were updated in Fig. 5A~B and Supplementary Figure 32~34. The electrochemical reconstruction of precursors are inevitable under the harsh CO₂ reduction conditions. As shown in Figure 2a, S27a, S28c and S29a, the catalysts have undergone a reduction and structural reconstruction and presented an enhanced Cu(111) peak after 50 h of electrolysis, but the Cu_L/Cu₂O catalyst shows protected Cu-O bonds proofed from the Raman signals assigned at 224, 425, 522 and 626 cm⁻¹.

Supplementary Figure 33 | (A and B) Operando SR-FTIR and (C and D) Raman measurements with potentials from -0.2 ~ -1.1 V and time during 20 min at -0.6 V vs. RHE for Cu_L/Cu₂O.

Supplementary Figure 34 | (A) Operando Raman measurements with various potentials from -0.2 ~ -1.1 V and time during 20 min at -0.6 V vs. RHE of Cu_P/Cu₂O.

The origins of Cu⁺ state can be summarized as follows:

(1) strain strengthening effect of residual Cl. The halogen ions play an important role in constructing Cu-based nanostructures and stabilizing cation Cu species (Nano Lett. 2019, 19, 3925–3932; Chem Electro Chem 2016, 3, 1012–1019; Angew. Chem. Int. Ed. 2022, 61,

e202116706). On the other hand, halogen ions in electrolyte that strongly adsorb on the surface of Cu catalysts could influence the surface charge property as well as the binding strength of certain intermediates (Catal. Commun. 2018, 114, 109–113; J. Electrochem. Soc. 2016, 163, F477–F484). For example, the addition of I⁻ ions into the CsHCO₃ electrolyte was proposed to contribute to the stabilization of Cu⁺ species under CO₂RR reaction by adsorbed iodine ions (ACS Catal. 2018, 8, 10012–10020). Similarly, the use of KCl electrolyte was also demonstrated to induce a biphasic Cu₂O-Cu catalyst and preserve Cu⁺ species under CO₂RR (Angew. Chem. Int. Ed. 2015, 54, 14701–14705).

(2) the existence of coordination defects and oxygen vacancies. For example, Li and co-workers prepared three Cu-based electrocatalysts with different oxidation states to study the valence state–activity relationship (J. Am. Chem. Soc. 2023, 145, 26133–26143). And, they suggest that the abundant oxygen vacancies can stabilize the oxidation state of Cu to improve the selectivity for C₂₊ products in CO₂RR. The electrocatalytic reduction process of Cu₂O to Cu in an aqueous solution involves the initial adsorption of H from H₂O onto the oxygen in Cu₂O. The energy released during the formation of bonds by oxygen-adsorbed water molecules was –2.06 eV, while the adsorption of the *COOH group at the vacancy releases an energy of –2.66 eV. The higher energy release indicates a more spontaneous reaction process, which suggests that oxygen vacancies regulate the electronic structure around Cu active sites to avoid their reduction. Besides, Zeng group successfully synthesized ZnO nanosheets with abundant oxygen vacancy, which restrained the self-reduction of ZnO nanosheets to the metallic state and improved the activity toward CO₂ electroreduction because of the faster charge transfer (Angew Chem Int Ed. 2018, 57, 6054–6059). Zheng group reported a partially reduced CuO_x nanodendrites with rich oxygen vacancies surface, which served as excellent Lewis base sites for enhanced CO₂ adsorption and investigated the significance of oxygen vacancies on the stability of copper oxides (Small Methods 2019, 3, 1800449).

(3) strong electronic interactions between carbon intermediate and Cu⁺. For instance, the electron configuration of CO is (1σ²2σ²3σ²4σ²1π⁴5σ²), in which the HOMO is the 5σ orbital and the LUMO is the 2π* orbital. When *CO bonds to the Cu surface, the 5σ and 2π* orbitals of *CO are each split into bonding and anti-bonding orbitals due to hybridization with the d-states of Cu. In this context, the 5σ orbital of *CO will donate electrons to the d-band of Cu to form a 5σ-d

donation, and the electrons in the d-band of Cu will be transported to $2\pi^*$ orbitals of *CO to form a d- $2\pi^*$ backdonation (J. Phys. Chem., 1964, 68, 2772–2777). The interaction between the d orbital of Cu and the frontier orbital of CO results in the formation of the bond. The electron back-donation weakens the internal connections between C and O because the $2\pi^*$ orbital is vacant and antibonding. Depending on the level of electron back-donation, the surface of Cu will adsorb CO molecules to varying degrees (ACS Catal. 2023, 13, 9222–9233; J. Phys.: Energy 2021, 3, 022001). Li group have shown differences of strong electronic interactions between carbon intermediate and Cu derived from Cu-NS, Cu₂O-NS and CuO-NS. As shown in Figure below, some of the electrons on Cu of Cu-NS are transferred to form an oxidized state, so that they can adsorb CO and OCCHO. Unlike Cu-NS, Cu on the surfaces of Cu₂O-NS and CuO-NS is already in an oxidized state, and Cu as an electron acceptor can effectively reduce the σ repulsion of *CO . The big aggregated electron clouds can be observed on the Cu–C bond, when CO is adsorbed (Figure i,j). At the same time, because CuO-NS has too strong of a binding energy for CO, the number of electrons transferred during C–C coupling is significantly higher than that required for Cu₂O-NS to realize C–C coupling (Figure l,m), which proved that Cu⁺ was the optimal oxidation state for adsorbing CO, further facilitating the C–C coupling (J. Am. Chem. Soc. 2023, 145, 26133–26143). And, Li and co-workers suggest that CO₂RR promotes the preservation of Cu⁺ species on catalyst surface (ACS Nano 2023, 17, 12884; Adv. Funct. Mater. 2023, 2310913). A significant *CO coverage exists on the Cu₂O/CuO surface during CO₂RR, which plays a crucial role in protecting and stabilizing the Cu⁺ species. This effect stabilizes the active Cu⁺ species during CO₂ electroreduction, which bind to Cu⁺ sites to cover the local catalyst surface.

Figure 6. Charge density difference of CO and OCCHO intermediate adsorption on (h,k) Cu-NS, (i,l) Cu₂O-NS, and (j,m) CuO-NS.

Fig. 1 | Catalytic mechanism of Cu_L/Cu₂O and free energy of intermediates: (A) Schematic diagram of rectifying interfaces in Mott-Schottky catalysts. (B) The adsorption of intermediates and C-C coupling reaction on Cu_P/Cu₂O. (C) The adsorption of intermediates and C-C or C₂-C coupling reaction on Cu_L/Cu₂O. (D) Free energy versus intermediate number of adsorbed *CO on the models of catalysts. (E) The formation energy of *CH₂CHO on Cu, Cu₂O, Cu_P/Cu₂O and Cu_L/Cu₂O. (F) The formation energy of *CH₂CO on Cu, Cu₂O, Cu_P/Cu₂O and Cu_L/Cu₂O.

In our manuscript, the discussion on catalytic mechanism based on DFT calculations mainly relied on four models of Cu, Cu₂O, Cu_P/Cu₂O and Cu_L/Cu₂O. In all these models, Cu₂O exhibit a

strongest linear relationship between the adsorption energy and *CO coverage, which implies the instability of Cu⁺ owing to the transfer and inject of number of electrons during C–C coupling. And, Cu_L/Cu₂O model displays a moderate affinity between carbon intermediate (*CO, *CH₂CHO and *CH₂CO) and catalyst surface. The related modifications are marked red in the revised manuscript.

[2]. In-situ Raman spectroscopy was employed to characterize the surface reaction intermediate. The spectra at lower Raman shift range (e.g., 200 ~ 1600 cm⁻¹) should be provided and discussed, which may help to probe the catalytic surface reconstruction. For example, the change of copper oxide band at 400~700 cm⁻¹ under different potentials.

Author response: We are so grateful to receive the reviewer's inspiring and professional suggestions about our work. Using Raman spectroscopy with laser excitation at 785 nm, we monitored the catalyst structural evolution in operando in the CO₂-flowed 0.1 M KHCO₃ electrolyte at an applied potential range from -0.2 ~ -1.1 V (Supplementary Figure 32 and 33). Cu₂O and Cl-Cu₂O samples exhibit characteristic signals at 224, 425, 522 and 626 cm⁻¹ attributed to the 2Γ⁻₁₂, 4Γ⁻₁₂, Γ⁺₂₅, and Γ⁻₁₂+Γ⁺₂₅ phonon modes, respectively (Solid State Commun. 1969, 7, 815–818; Phys. Rev. B 1975, 12, 1377–1394). These Raman signals for the Cu₂O precursor disappear gradually in the potential range from -0.2 to -0.7 V vs RHE, corresponding to the full Cu₂O reduction to metallic Cu. In sharp contrast, the Cl-Cu₂O retains the characteristic Raman modes (2Γ⁻₁₂ and Γ⁻₁₂+Γ⁺₂₅ mode) until the potential > -0.7 V, suggesting that the low-coordinated Cu_L/Cu₂O can protect Cu⁺ species against reduction owing to the combination of environmental and structural factors. In addition, Raman measurement displays a prominent emerged peak at 2000~2100 cm⁻¹, which corresponds to the νCO vibrational mode of *CO. Overall, Cu_L/Cu₂O catalyst has stronger adsorption capacity for *CO attributed to the restricted rotation and stretching vibration peaks of adsorbed *CO and Cu–CO bond around 310 and 358 cm⁻¹, respectively. In particular, the *CO coverage is closely related to the intensity ratio of the ν(Cu–CO) to the ν(*CO). Obviously, the intensity ratio of the ν(Cu–CO) to the ν(*CO) of Cu_L/Cu₂O is greater than Cu_L/Cu₂O, indicating that *CO coverage is higher and beneficial to the oxidation state of Cu⁺. The peaks at 1024 and 1069 cm⁻¹ are attributed to ν₁(C–O) of HCO₃⁻ and ν₂(C–O) of CO₃²⁻,

respectively. As the potential increases, the CO_3^{2-} accumulates and disappear on the surface of catalyst, which presents the dissolution and reduction of CO_2 . The related modifications are marked red in the revised manuscript.

Supplementary Figure 33 | (C and D) Raman measurements with potentials from -0.2 ~ -1.1 V and time during 20 min at -0.6 V vs. RHE for $\text{Cu}_L/\text{Cu}_2\text{O}$.

Supplementary Figure 34 | (A) Operando Raman measurements with various potentials from -0.2 ~ -1.1 V and time during 20 min at -0.6 V vs. RHE of $\text{Cu}_P/\text{Cu}_2\text{O}$.

[3]. In page 6 line 3~4, CH_2CHO is claimed as a key intermediate for ethanol. However, it is also regarded as an intermediate of ethylene according to some literature (Adv. Mater. 2021, 2005798).

Author response: Thanks for the reviewer to point out our carelessness. I apologize for the imprecise expression and revise the relevant part of manuscript. In page 6 line 3~4, the described “key intermediate for ethanol” is substituted by “branching intermediate for ethanol or ethylene”.

The *CH_2CHO intermediate at sixth proton-electron transfer step will be hydrogenated for $^*O\&C_2H_4$ or $^*CH_2CH_2O^*$ intermediates. The related modifications are marked red in the revised manuscript.

Figure. An overview of mechanistic proposals of CO₂ and CO electroreduction on Cu catalysts (Adv.Mater.2021, 33, 2005798).

Figure. Possible mechanistic pathways of CO₂ reduction to C₂H₄ and C₂H₅OH (ACS Energy Lett. 2021, 6, 694–706).

[4]. The coexistence of XRD patterns of Cu₂O and Cu cannot prove “the Mott–Schottky interface of metal/metal oxides” (Page 7, line 3-4). The expression is inaccurate because the physical mixing has the same result.

Author response: Thanks for the reviewer to point out our inaccurate expression. In page 7 line 3~4, the described “which proof the Mott–Schottky interface of metal/metal oxides” is substituted by “which proof the coexistence of Cu⁺ and Cu⁰”. The related modifications are marked red in the revised manuscript.

[5]. The particle sizes of Cu_L/Cu₂O and Cu_P/Cu₂O were different. The possible size effect should be excluded to support the discussion on catalytic mechanism.

Author response: We appreciate the reviewer to point this problem out for us. According to the reviewer's suggestion, the Cl-Cu₂O powders with different particle sizes were prepared by modulating the inputs of reactant and reaction time. The morphological structure and electrochemical properties of samples were records in Supplementary Figure 29 and 30. The

excessively small particle size is unfavorable for the generation of oxygenated products due to the metallization of particles. In addition, the activity of hydrogen evolution reaction will be enhanced on the surface of catalysts with smaller particle sizes (J. Am. Chem. Soc. 2014, 136, 19, 6978–6986). Nevertheless, the selectivity of C_{2+} alcohols are gradually increased on gas diffusion electrode loaded with larger size particles, which benefits from the stability of low-coordinated Cu/Cu₂O. However, the selectivity of C_{2+} products steep decline owing to the aggregation of nanoparticles. In brief, the appropriate particle size is contributed to the stabilization of enhanced rectifying interface of low-coordinated Cu_L/Cu₂O Mott–Schottky catalyst. The related modifications are marked red in the revised manuscript.

Supplementary Figure 29 | SEM images of Cl-Cu₂O with different particle size: (A) ~30 nm, (B) ~150 nm, (C) ~500 nm, (D) ~1 μm

Supplementary Figure 30 | FE of products derived from Cl-Cu₂O with different particle size: (A) ~ 30 nm, (B) ~ 150 nm, (C) ~ 500 nm, (D) $\sim 1\mu$ m in 1 M KOH electrolyte.

Figure. Particle size dependence of (a) the composition of gaseous reaction products (balance is CO₂) during catalytic CO₂ electroreduction over Cu NPs, (b) the faradaic selectivities of reaction products during the CO₂ electroreduction on Cu NPs. Lines are guides to the eye. Conditions: 0.1 M KHCO₃, $E = -1.1$ V/RHE, 25 °C (*J. Am. Chem. Soc.* 2014, 136, 19, 6978–6986).

[6]. I'm a little confused about the work functions estimated by KPFM (Figure 2E, S12). More discussion on this point is suggested.

Response: Thank you for your valuable comments. The topography and surface potential map of samples were characterized simultaneously by using KPFM. The schematic diagram of the KPFM system is shown in Figure (*J. Mater. Chem. A*, 2020, 8, 8586-8592). The topographies and surface potentials were measured by KPFM based on dynamic force microscopy principles to give work functions of the samples. The work function ϕ_{sample} was further calculated from the surface potential according to equation:

$$\phi_{\text{Sample}} = \phi_{\text{Tip}} + qV_{\text{CPD}}$$

$$V_{\text{CPD}} = V_{\text{CPD-sample}} - V_{\text{CPD-substrate}}$$

$$\phi_{\text{Sample}} = \phi_{\text{substrate}} + qV_{\text{CPD}}$$

where $V_{\text{CPD-sample}}$, $V_{\text{CPD-substrate}}$ are the contact potential difference of sample and substrate measured by KPFM in volts. ϕ_{Tip} , ϕ_{sample} and $\phi_{\text{substrate}}$ are the work functions of tip, sample and substrate in eV, respectively and q is the electronic charge. The work functions of tip can be calculated by measuring gold samples by KPFM.

Figure. The schematic diagram of the KPFM system (J. Mater. Chem. A, 2020, 8, 8586-8592).

In this manuscript, the substrate is SiO_2 (WF=5.05 eV) and the standard gold disk with a work function of 5.3 eV is used to calibrate the probe. Figure 2E and Supplementary Figure 3 exhibit the contact potential difference (CPD) profiles for each material, where the average surface potential can be estimated to be 0.042, 0.035, 0.024 and 0.037 V for Cu_2O , Cl- Cu_2O , $\text{Cu}_p/\text{Cu}_2\text{O}$ and $\text{Cu}_l/\text{Cu}_2\text{O}$, respectively. From the CPD profiles, the work function of Cu_2O , Cl- Cu_2O , $\text{Cu}_p/\text{Cu}_2\text{O}$ and $\text{Cu}_l/\text{Cu}_2\text{O}$ samples are determined to be 5.1, 5.08, 5.03 and 5.01 eV, respectively. These values are in accordance with the trends of work function calculated by UPS. The related modifications are marked red in the revised manuscript.

[7]. Cu 2p and LMM spectra should be analyzed through peak deconvolution (Surf. Interface Anal. 2017, 49, 1325), which will help to identify the chemical state of surface Cu. From the shakeup peak in Figure 3E, it seems that the $\text{Cu}_p/\text{Cu}_2\text{O}$ had a notable amount of Cu^{2+} species, whereas the $\text{Cu}_l/\text{Cu}_2\text{O}$ did not.

Response: We thank the reviewer for the professional suggestion, which is very important to

deeply to identify the chemical state of surface Cu. For the question of a different Cu 2p spectra of $\text{Cu}_\text{P}/\text{Cu}_2\text{O}$ and $\text{Cu}_\text{L}/\text{Cu}_2\text{O}$, we have repeated a XPS experiment to verify the result of manuscript as shown below. The satellite peaks assigned to the characteristic signal of Cu^{2+} is clearly detected in $\text{Cu}_\text{P}/\text{Cu}_2\text{O}$ catalyst and slightly present in $\text{Cu}_\text{L}/\text{Cu}_2\text{O}$ catalyst, which is consistent with the Cu 2p spectra of manuscript. Based on the results of the experiments (Supplementary Figure 26), we hypothesize that $\text{Cu}_\text{P}/\text{Cu}_2\text{O}$ catalyst is more easily oxidized when exposed to air owing to a greater metallization during electrochemical reaction, but the low-coordinated $\text{Cu}_\text{L}/\text{Cu}_2\text{O}$ has a higher CER activity due to the presence of oxygen vacancies and Cl for stabilizing the interface of Cu/ Cu_2O (Chem Catalysis 2023, 3, 100562). The related modifications are marked red in the revised manuscript.

Figure. Repeated XPS experiment. (A) Cu 2p and (B) Cu LMM Auger spectra of $\text{Cu}_\text{P}/\text{Cu}_2\text{O}$ and $\text{Cu}_\text{L}/\text{Cu}_2\text{O}$.

To identify clearly the chemical state of surface Cu, according to the suggestion of reviewer, Cu 2p and LMM spectra have been analyzed through peak deconvolution, which are revised in Fig. 3, Supplementary Figure 2, 4, 16 and 24. Based on the relevant references, the Cu 2p can be split into two characteristic signals for Cu^{2+} and Cu^0/Cu^+ , and Cu LMM is used for the fine distinction of Cu^{2+} , Cu^0 and Cu^+ (Surf. Interface Anal. 2017, 49, 1325; J. Am. Chem. Soc. 2023, 145, 26133–26143).

Figure. (A) Cu 2p_{3/2} and (B) Cu L₃M_{4,5}M_{4,5} spectra from a sample of as received cold sprayed copper on a steel substrate (Surf. Interface Anal. 2017, 49, 1325). (C) Cu 2p and (D) AES spectrum of Cu LM2 (J. Am. Chem. Soc. 2023, 145, 26133–26143).

Fig. 3 | Electron structures and coordination of $\text{Cu}_I/\text{Cu}_2\text{O}$: (A) O 1s XPS spectra of $\text{Cu}_I/\text{Cu}_2\text{O}$ with or without Ar^+ etching. (B) EPR spectra of $\text{Cu}_I/\text{Cu}_2\text{O}$ and $\text{Cu}_P/\text{Cu}_2\text{O}$. (C) Cl 2p, (D) Cu 2p and (E) Cu LMM Auger spectra of $\text{Cu}_P/\text{Cu}_2\text{O}$ and $\text{Cu}_I/\text{Cu}_2\text{O}$. (F) The normalized Cu K-edge EXAFS spectra and (G) fourier transform k^3 -weighted EXAFS for Cu foil, Cu_2O , CuO and $\text{Cu}_I/\text{Cu}_2\text{O}$. (H) EXAFS fitting curve of $\text{Cu}_I/\text{Cu}_2\text{O}$. (I) wavelet transform for the k^3 -weighted EXAFS for Cu foil, Cu_2O and $\text{Cu}_I/\text{Cu}_2\text{O}$.

In contrast with Cu 2p spectra of the $\text{Cu}_P/\text{Cu}_2\text{O}$ species, the Cu 2p 3/2 peak of $\text{Cu}_I/\text{Cu}_2\text{O}$ shows a strong characteristic signal for Cu^0/Cu^+ at 932.3 eV, accompanied by weak satellite peaks

(Fig. 3D). The AES of $\text{Cu}_\text{I}/\text{Cu}_2\text{O}$ and $\text{Cu}_\text{L}/\text{Cu}_2\text{O}$ were analyzed by Cu LMM Auger spectra to prove the valence of Cu more clearly (Fig. 3E). The kinetic energy of the Auger electron transition of $\text{Cu}_\text{L}/\text{Cu}_2\text{O}$ split for Cu^+ , Cu^{2+} and Cu^0 is 916.4, 917.2 and 918.3 eV, respectively. This further proves the successful synthesis of $\text{Cu}_\text{I}/\text{Cu}_2\text{O}$.

Supplementary Figure 2 | Nitrogen adsorption and adsorption curves and High-resolution XPS. (A) Nitrogen adsorption and adsorption curves, (B) High-resolution Cu 2p, (C) Cu LMM and (D) O 1s of as-synthesized Cu_2O .

Supplementary Figure 4 | The XPS spectra of Cl-Cu₂O. (A-C) High-resolution O 1s, (D-F) Cu 2p, (G-I) Cu LMM and (J-L) Cl 2p XPS patterns of Cl-Cu₂O with or without Ar⁺ etching.

Supplementary Figure 24 | (A-E) Cu 2p, (B, F) O 1s, (C, G) Cl 2p and (D, H) Cu LMM XPS spectra of as-synthesized $\text{NH}_4\text{Cl-Cu}_2\text{O}$ with or without Ar^+ etching.

The abundant oxygen vacancies are detected on the surface of Cl-Cu₂O, whereas the signals gradually disappear by increasing Ar⁺ etching. In addition, it can be observed that some Cu of Cl-Cu₂O has been reduced to low valence states (Supplementary Figure 4 G-I), which also suggests that there may be some coordination defects owing to the introduce of Cl, leading to changes in the valence states of Cu around the defects.

Supplementary Figure 16 | (A and B) High-resolution Cu 2p and (C and D) Cu LMM XPS patterns of Cu_I/Cu₂O after Ar⁺ etching.

[8]. Error bars should be added in the CO₂ reduction performance tests.

Response: Thank you for your valuable comments. In order to ensure the reality and accuracy of results, we repeated the experiments three times to verify the electrochemical test results, and the relevant product distributions have been modified into column chart with error bars. The related modifications are marked red in the revised manuscript.

Fig. 4 | Electrochemical performance of CO₂ reduction: (A) LSV curves acquired in flow cell using 1 M KOH as electrolyte. (B and C) Faradic efficiencies of products at various j_{total} using Cu_P/Cu₂O (B) and Cu_L/Cu₂O (C) catalysts. (D) C₂₊ and C₂₊ alcohols partial current density of Cu_P/Cu₂O and Cu_L/Cu₂O catalysts. (E and F) Energy efficiencies (E) and production rate (F) of C₂₊ alcohols on Cu_P/Cu₂O and Cu_L/Cu₂O catalysts. (G) Summary of EE_{C₂₊ alcohols} versus FE_{C₂₊ alcohols} of this work with other catalysts in flow cell. (H) Electrochemical stability test at the j_{total} of 200 mA cm⁻² using the Cu_L/Cu₂O catalysts.

Supplementary Figure 25 | (A) FE and (C) partial current density of NH₄Cl-Cu₂O in 1 M KOH electrolyte. (B) FE and (D) partial current density of Cu_p/Cu₂O in 1 M KOH electrolyte with 3 M KCl.

Supplementary Figure 30 | FE of products derived from Cl-Cu₂O with different particle size: (A) ~30 nm, (B) ~150 nm, (C) ~500 nm, (D) ~1 μm in 1 M KOH electrolyte.

[9]. Why would the Cl⁻ ions in the electrolyte adsorb on the negatively-charged electrode surface in CO₂ reduction (Page 14-15)?

Author response: We really appreciate the reviewer for the professional and meaningful suggestion regarding additional evidence to support our claimed catalytic mechanism in the manuscript. The promotional effects of halide ions ($X^- = Cl^-$, Br^- , and I^-) in CO₂ reduction have yet to be well investigated (ACS Catal. 2017, 7, 5112–5120; J. CO₂ Util. 2013, 1, 43–49; Electrochim. Acta 2010, 56, 381–386; Angew. Chem., Int. Ed. 2021, 60, 14329–14333; J. Am. Chem. Soc. 2023, 145, 8714–8725). These weakly solvated anions can specifically adsorb on the inner Helmholtz plane (IHP) of the catalyst surface, which not only confine CO₂ and facilitate electron transfer from the electrode to CO₂ via the X_{ad}^- -C bond but also improve the *CO adsorption for favorable CO–CO coupling (J. Am. Chem. Soc. 2023, 145, 8714–8725). Ogura and workers have studied the effect of anions for CO₂ reduction and suggest that the specific adsorption

of halide anions onto copper electrode can build a double layer at the metal/solution interface (J. CO2 Util. 2013, 1, 43–49). Metallic ions with strongly bound water molecules interact with the metal electrode by electrostatic forces, while halide anions with weakly bound water molecules can remove a part of water and form a direct chemical bond with the metal electrode. In the latter case, a surface concentration of anions is in excess of that provided with pure electrostatic forces. This consequence is called specific adsorption in general. The specific adsorption is ascribed not to the electrostatic attraction but to a strong chemical affinity of the anion for metal. The halide anions and solvated cations are located on the inner and outer Helmholtz planes, respectively, as shown in Figure below. Specifically adsorbed anions can block the adsorption of protons, and the overpotential for the hydrogen evolution is enhanced. Thus, the proton electroreduction competitive to the reduction of CO₂ can be suppressed. On the other hand, the reduction of CO₂ is expedited by the existence of specifically adsorbed anions as described later. For this reason, the hydrogen evolution may be effectively controlled even in acidic solution. The related modifications are marked red in the revised manuscript.

Table. Current efficiencies of products from the electroreduction of CO₂ at a Cu electrode at 5 mA/cm² in various solutions at 19 °C (J. CO2 Util. 2013, 1, 43–49).

Electrolyte (M)	pH ^a	Potential vs NHE	Faraday efficiency (%)							
			CH ₄	C ₂ H ₄	EtOH	n-PrOH	CO	HCOO ⁻	H ₂	Total
KHCO ₃	0.1	6.8	29.4	30.1	6.9	3.0	2.0	9.7	10.9	92.0
KCl	0.1	5.9	11.5	47.8	21.9	3.6	2.5	6.6	5.9	99.8
	0.5	–	14.5	38.2	–	–	3.0	17.9	12.5	–
KClO ₄	0.1	5.9	10.2	48.1	15.5	4.2	2.4	8.9	6.7	96.0
K ₂ SO ₄	0.1	5.8	12.3	46.0	18.2	4.0	2.1	8.1	8.7	99.4
K ₂ HPO ₄	0.1	6.5	17.0	1.8	0.7	tr	1.3	5.3	72.4	98.5
	0.5	7.0	6.6	1.0	0.6	0.0	1.0	4.2	83.3	96.7

^a pH values were measured for bulk solutions after electrolyses.

Figure. CO₂ attraction by electric double layer and formation of formate radical by inner-sphere mechanism (J. CO2 Util. 2013, 1, 43–49).

Figure. Schematic of specifically adsorbed anions (Br^-), approach of CO_2 , and subsequent electrochemical reduction (Electrochim. Acta 2010, 56, 381–386).

Figure. Production rates of (a) C_2H_4 , (b) CH_4 , (c) CO , (d) $HCOO^-$, (e) C_2H_5OH and (f) $n-C_3H_7OH$ as a function of applied potential for O_2 -plasma-treated Cu foils in different electrolytes, as well as for an electropolished metallic Cu foil (open squares) in pure $KHCO_3$ solution after 1 h of CO_2 electroreduction.

[10]. Some intermediates were detected in the operando FT-IR. However, the catalytic mechanism was not well discussed based on these intermediates. The operando FT-IR of Cu_p/Cu_2O should also be performed to clarify the catalytic mechanism. Besides, the electrolyte in operando FTIR and Raman tests was different from that in CO_2 performance tests. As the electrolyte was found to have a critical role in CO_2 reduction, the effect of electrolyte should be considered.

Author response: We thank the reviewer for this professional question. The Operando SR-FTIR experiments have been updated in Fig. 5A and Supplementary Figure 32. The reference spectrum is taken at open-circuit potential and additional spectra are provided from -0.2 ~ -1.1 V vs. RHE. After CO₂ is bubbled through the solution, two bands appear at 1663 and 1280 cm⁻¹, corresponding to the C=O and C–OH stretch of COOH. These peaks indicate that the COOH* intermediate is present during the CO₂ reduction on the Cu_L/Cu₂O. Subsequently, the C≡O stretch band in the 2000–2100 cm⁻¹ is attributable to linearly bonded in an a-top geometry on the edge sites (CO_L). In stark contrast to the behavior of the CO_L band, the band in the ≈1800–1900 cm⁻¹ region caused by CO bridge (CO_B) adsorbed on two Cu atoms exhibits a remarkable degree of hysteresis. The red shift of CO_B band at more negative potentials is caused by the Stark tuning effect. Simultaneously with the CO-related band, a band at 1305 cm⁻¹ associated with *COH stretching grows and *COOH disappears in the spectra, which means the performance of CO₂ reduction. The absence of the bands at 1589 cm⁻¹ in the transmission spectra points out the possibility that the *OCCOH coupling reaction during the reduction of CO₂ on Cu_L/Cu₂O. Combined with time-independent spectral analysis, the stretch of OCH₂CH₃ assigned at 1335 cm⁻¹ implies a pathway for the production of C₂₊ alcohols, which displayed in Fig. 5F. However, in contrast to Cu_L/Cu₂O catalyst, some new signals at 1706, 1543 and 1399 cm⁻¹ corresponded to the stretch of *CHO, *OCCO and *OCHO intermediates are emerged in spectra of Cu_P/Cu₂O, which certifies the coupling mode of OC-CO and possible formate product. These conclusions are consistent with the results of products counted from Cu_P/Cu₂O and Cu_L/Cu₂O.

The effect of electrolytes have been conducted in Supplementary Figure 31. As the pH of electrolyte increases, the FE of C₂₊ products on Cu_L/Cu₂O catalyst gains a climb up to ~90%, where the selectivity of C₂₊ alcohols reaches a maximum value in 1 M KOH and decline slightly in more basic electrolyte. In conclusion, it undergoes a electrochemical reduction reaction of CO₂ with synergistic effect of CO₂ coverage and catalyst in pH-compatible electrolyte and thermodynamics-mediated competitive reaction in strong alkaline electrolyte. Besides, the Cl element shows a positive effect for improving the selectivity of C₂₊ products, but the sensitivity of ethylene is more forceful than C₂₊ alcohols. The related modifications are marked red in the revised manuscript.

Fig. 5 | Operando experiments and DFT calculations: (A) operando FTIR and (B) Raman tests of $\text{Cu}_L/\text{Cu}_2\text{O}$ using 0.1 M KHCO_3 as electrolyte. (C and D) Bader charge analysis and bond length of $\text{Cu}_P/\text{Cu}_2\text{O}$ and $\text{Cu}_L/\text{Cu}_2\text{O}$, respectively. (E) Gibbs free energy of $^*\text{CH}_2\text{COCO}$ intermediates for C_3 products and (F) C-C coupling to C_2 products (ethanol and ethylene) on $\text{Cu}_P/\text{Cu}_2\text{O}$ and $\text{Cu}_L/\text{Cu}_2\text{O}$ catalysts.

Supplementary Figure 33 | (A and B) Operando SR-FTIR and (C and D) Raman measurements with potentials from -0.2 ~ -1.1 V and time during 20 min at -0.6 V vs. RHE for Cu_L/Cu₂O.

Supplementary Figure 32 | (A) Operando SR-FTIR measurements of Cu_P/Cu₂O with various potentials from -0.2 ~ -1.1 V.

Supplementary Figure 31 | FE of products in 0.1 (A), 0.5 (B), 1 (C) M KHCO_3 , 2 M KOH (D) and 3 M KCl (E) electrolyte, and (F) statistical bar chart of C_{2+} alcohols and ethylene for $\text{Cu}_L/\text{Cu}_2\text{O}$.

[11]. Some Figures are too blurring to be seen clearly. Besides, there are many clerical errors, for example, Figure 3B legend, $\text{Cu}_L/\text{Cu}_2\text{O}$ catalyst in Page 16 line 12, sample abbreviation in Figure S2a, standard patterns of Cu and Cu_2O in Figure S27a.

Author response: We thank the reviewer for pointing out our mistake. We have updated some blurred images, such as Supplementary Figure 26. And, We have checked carefully the details in manuscript to avoid some clerical errors.

Supplementary Figure 26 | (A) XRD patterns of $\text{Cu}_P/\text{Cu}_2\text{O}$ and $\text{Cu}_I/\text{Cu}_2\text{O}$ catalysts after CER. SEM images of $\text{Cu}_P/\text{Cu}_2\text{O}$ (B), $\text{Cu}_I/\text{Cu}_2\text{O}$ (C) and $\text{NH}_4\text{Cl-Cu}_2\text{O}$ (D) after CER.

We appreciate the reviewer again for your helpful comments on the manuscript in the hectic schedules, and we hope that the appropriate revision will improve the article's quality and provide readers with a good experience. Best wish for you!

Reviewer #3: In this work, Zhang et.al use an electrochemical reconstruction strategy on chlorine doped cuprous oxide to maximize C₂₊ alcohols production and report a faradaic efficiency of 68 % at 200 mA/cm². While the catalyst design strategy is quite interesting, the work lacks significant novelty and are inconsistent in explanation at various places which make it unsuitable for publication in Nature Communications journal. This work may be more suitable for a more catalyst focused journal.

Author response: Thanks again for the reviewer's precious time and professional suggestions, which are very important as an insightful guidance for us to further improve the quality of our manuscript. We sincerely hope that our revisions according to the reviewer's suggestions can now meet the high standards of Nature Communications. We really appreciate your precious time on reviewing our revised manuscript.

In this work, we have focused on the effect of the rectifying interface of Cu/Cu₂O for selective C₂₊ alcohols, which will boost the specific adsorption of intermediates at the enhanced rectifying interface by adjusting the coordination environment of Cu sites in Cu/Cu₂O. Catalysts with different rectifying interface effects of Cu/Cu₂O were prepared by electrochemical reconstruction of Cu precursors. Experiments and studies certify that the Cu_L/Cu₂O catalyst with low-coordinated Cu sites can lead to the enhanced rectifying interfaces, and thus induce the asymmetric electronic perturbation and faster electron exchange for boosting C-C coupling and bonding oxyhydrocarbons toward nucleophilic reaction process of *H₂CCO-CO. In contrast, the Cu_P/Cu₂O catalyst with a weak rectifying interface is unable to stabilize the adsorbed oxyhydrocarbons intermediates, which exhibits a high ethylene selectivity. And, there are few studied on rectifying interface of Cu/Cu₂O, especially the roles of rectifying interface for the selectivity between ethylene and C₂₊ alcohols. In brief, there are three highlights for publishing the manuscript in Nature Communications.

1. the delicate control of rectifying interface of Cu/Cu₂O for bonding *CO and oxyhydrocarbons increase the selectivity of C₂₊ alcohols

We suggest that the enhanced rectifying interface will facilitate *CO coverage on the low-coordinated Cu sites of oxide-derived Cu and further hydrogenation to *COH, which are essential for the coupling of OC-COH. The electronic perturbation of Cu and fast electron exchange in the enhanced rectifying interface of Cu/Cu₂O can boost C-C coupling and bond oxyhydrocarbons

toward nucleophilic reaction process of $*\text{H}_2\text{CCO}-\text{CO}$ by electrostatic tension. The electron-rich regions facilitate the generation of ethanol due to the strengthened adsorption of $*\text{C}_2\text{H}_2\text{O}$ and $*\text{C}_2\text{H}_3\text{O}$ under the electric field. More critically, the fast electron exchange and enhanced intermediates adsorption at synergistic rectifying interface regions due to the low coordination and asymmetric electron aggregation inside $\text{Cu}_\text{L}/\text{Cu}_2\text{O}$ catalyst. The excellent conductivity and charge difference of Cu^+/Cu^0 boost the nucleophilic or electrophilic addition reaction process for C-C or C_2-C coupling.

Many articles that Cu-based catalysts are applied to the electrochemical reduction reaction of CO_2 for the C_{2+} products are published on Nature communications. For example, Han group reported a $\text{Ag}_x\text{S}-\text{Cu}_2\text{O}/\text{Cu}$ electrocatalyst, which exhibits a methanol faradaic efficiency of 67.4% with a current density as high as 122.7 mAcm^{-2} owing to the optical electronic structure and morphology of the catalysts to favor methanol pathway and the suppressed hydrogen evolution reaction with the introduce of anion S and cation Ag (Nat. Commun. 2022, 13, 1965). In addition, Wu and co-worker report a cascade AgCu single-atom and nanoparticle electrocatalyst, in which Ag nanoparticles produce CO and AgCu single-atom alloys promote C-C coupling kinetics. As a result, AgCu SANP catalyst presents a C_{2+} faradaic efficiency of $94\pm 4\%$ with the $\sim 720 \text{ mA cm}^{-2}$ working current density at -0.65V in a flow cell with alkaline electrolyte. The Ag single-atom doping of Cu nanoparticles will increase the adsorption energy of $*\text{CO}$ on Cu sites due to the asymmetric bonding of the Cu atom to the adjacent Ag atom with a compressive strain. Moreover, other articles achieve the excellent C_{2+} performance by Cu-based catalysts as follows:

- (1) Nat. Commun. 2021, 12, 586;
- (2) Nat. Commun. 2023, 14, 3075;
- (3) Nat. Commun. 2023, 14, 2823;
- (4) Nat. Commun. 2023, 14, 6164;
- (5) Nat. Commun. 2023, 14, 3382;
- (6) Nat. Commun. 2023, 14, 7833;
- (7) Nat. Commun. 2023, 14, 4615;
- (8) Nat. Commun. 2023, 14, 335;
- (9) Nat. Commun. 2023, 14, 5245;
- (10) Nat. Commun. 2023, 14, 6849;

- (11) Adv. Sci.2022,9, 2200454;
 (12) Nat. Catal. 2020, 3, 75-82;
 (13) J. Am. Chem. Soc. 2020, 142, 6878-6883;
 (14) Nat. Commun. 2020, 11, 3622;
 (15) Angew. Chem. Int. Ed. 2022, e202212640;

2. Excellent performance and high C₂₊ alcohols selectivity on the enhanced rectifying interface of Cu₁/Cu₂O

In this manuscript, Cu₁/Cu₂O catalyst achieves superior C₂₊ alcohols selectivity: faradic efficiency of 64.2%, energy efficiency of 39.3% and a stability of ~ 50 h (FE_{C₂₊ alcohols} > 50%, j_{total}=200 mA cm⁻²).

A detailed summary of the performance of other catalysts for ethanol could be found in Figure and Table.

Figure. A comparison of C₂₊ alcohols for various electrocatalysts in flow cell electrolyzer.

Table. A comparison of C₂₊ alcohols for various electrocatalysts corresponds to Figure 1.

Catalysts	Electrolyzer	Electrolytes	Potential (V vs. RHE)	J _{total} (mA·cm ⁻²)	J _{ethanol} (mA·cm ⁻²)	EE _{alcohols}	Reference
Cu ₁ /Cu ₂ O	Flow cell	1 M KOH	-0.66	-200	-113	39.32	This work
K-F-Cu-CO ₂	Flow cell	1 M KOH	-0.53	-800	-423	36.45	Adv. Mater. 2022, 34, 2204476

BaO/Cu	Flow cell	1 M KOH	-0.75	-400	-204	35.06	Nat. Catal. 2022, 5, 1081–1088
Cu-DS	H-cell	0.1 M KHCO ₃	-1.08	-32	-16	34.96	Joule 2021, 5, 429–440
	Flow cell	1 M KOH	-0.95	-200	-50	34.97	
R-Cu-C	Flow cell	1 M KOH	-0.8	-47.4	15.4	33.18	Chem Catal. 2023, 3, 100512
N-C/Cu	Flow cell	1 M KOH	-0.68	-300	-156	31.86	Nat. Energy 2020, 5, 478–486
NGQ/Cu-nr	Flow cell	1 M KOH	-0.9	-281.2	-126.5	27.8	Angew. Chem. Int. Ed. 2020, 59, 16459–16464
CuZn alloy	Flow cell	1 M KOH	-0.68	-200	-84	27.43	Angew. Chem. Int. Ed. 2019, 58, 15036–15040
wr-Cu	Flow cell	1 M KOH	-0.91	-800	-328	26.43	Chem. Sci. 2023, 14, 310–316
FeTPP[Cl]/Cu	Flow cell	1 M KHCO ₃	-0.82	-302	-124	24.56	Nat. Catal. 2020, 3, 75–82
Hex-2Cu-O	H-cell	0.1 M KHCO ₃	-1.2	-8.5	-2.76	23.76	Nat. Commun. 2022, 13, 5122
	Flow cell	1 M KOH	-0.66	-200	-26	23.25	
Cu-CuI	Flow cell	1 M KOH	-1	-900	-261	18.35	Angew. Chem. Int. Ed. 2021, 60, 14329–14333
Cu(OH) ₂ -D	Flow cell	1 M KOH	-0.54	-250	-42.5	17.99	Angew. Chem. Int. Ed. 2021, 60, 4879–4885
Li _{2-x} CuO ₂ -10	Flow cell	1 M KOH	-0.85	-220	-59.4	16.21	Small 2022, 18, 2106433
F-Cu	Flow cell	2.5 M KOH	-0.54	-400	-68	12.86	Nat. Catal. 2020, 3, 478–487
Cu ₂ S-Cu-V	Flow cell	1 M KOH	-0.95	-400	-92	9.45	Nat. Catal. 2018, 1, 421–428
Cu-CIPH	Flow cell	7 M KOH	-0.91	-1170	269.1	7.3	Science 2020, 367, 661–666
Cu/NPC-800	H-cell	0.2 M KHCO ₃	-1.05	-12.13	8.2	36.61	Green Chem. 2020, 22, 71–84
Cu ₂ O-ZnO	H-cell	0.5 M KHCO ₃	-0.9	-	-	31.47	J. Power Sources, 2023, 556, 232468
Cu ₃ Ag ₁	H-cell	0.5 M KHCO ₃	-0.95	-39.7	-17.2	27.64	Adv. Energy Mater. 2020, 10, 2001987
CeO ₂ -Cu	H-cell	0.1 M KHCO ₃	-0.6	45.3	13.14	20.45	ACS Materials Lett. 2022, 4, 1999–2008
Cu_I	H-cell	0.1 M KHCO ₃	-0.9	-38	-10.5	19.76	Angew. Chem. Int. Ed. 2019, 58, 17047–17053
CuO-ZnO ₁₀	H-cell	0.5 M KHCO ₃	-0.8	-5	-3.6	16.26	Electrochimica Acta 2021, 392, 138988
od-Pd ₉ Cu ₉₁	H-cell	0.5 M KHCO ₃	-0.65	-8.4	-1.3	12.54	Green Chem. 2020, 22, 6497–6509

3. the clear mechanism exploration of low-coordinated Cu_L/Cu₂O on stabilizing *CO and oxyhydrocarbons intermediates

Compared with pure Cu_P/Cu₂O catalyst, the low-coordinated Cu_L/Cu₂O possess the advantages as follows: (1) the fast electron exchange and enhanced intermediates adsorption at synergistic rectifying interface regions due to the low coordination and asymmetric electron aggregation inside Cu_L/Cu₂O catalyst. The excellent conductivity and charge difference of Cu⁺/Cu⁰ boost the nucleophilic or electrophilic addition reaction process for C-C or C₂-C coupling. Second, the fast electron transfer from Cu to Cu⁺ at the rectifying interface can induce lattice shrinkage and inhibit oxygen atoms loss, which take responsibility for stabilizing chemical valence of Cu⁺ with residual Cl atoms. The clean Cu/Cu₂O regions ensures the smooth absorption and desorption as well as coupling behavior of bonding oxyhydrocarbons intermediates. Third, abundant defects originated from oxygen vacancies and Cl doping supply rich free electron and landing sites for adsorbed intermediates. Density functional theory calculations and operando SR-FTIR and Raman experiments decipher that Cu_L/Cu₂O can enhance the coverage of *CO and adsorption of *CH₂CO and CH₂CHO toward the formation of C₂₊ alcohols.

In order to improve the quality of the manuscript, we have made meticulous revisions based on the professional suggestions of reviewers to reduce the flaws of manuscript. And, we hope that our revisions will improve the interest of readers and up to the standard of Nature communications. We would be deeply grateful for your kindest assistance in the course of the reviewing process of this manuscript.

[1]. The authors use Ag/AgCl reference electrode in 1 M KOH electrolyte for CO₂ electrolysis for the prepared Cu catalysts. There have been several reports in the community recommending researchers not to use Ag/AgCl reference electrodes in alkaline media. Hence, all tests must be reported using a Hg/HgO or Hydroflex reference electrode.

Response: Thank you for your valuable comments. We have repeated all tests using a Hg/HgO electrode as reference electrode. For example, Fig. 4 and Supplementary Figure 20, 21, 25, 30, etc. have been updated after using Hg/HgO electrode. The related modifications are marked red in the revised manuscript.

Figure. Optical picture of flow cell with Hg/HgO electrode as reference electrode.

Supplementary Figure 20 | (A) Tafel curves and (B) EIS spectra of $\text{Cu}_p/\text{Cu}_2\text{O}$ and $\text{Cu}_l/\text{Cu}_2\text{O}$.

Supplementary Figure 21 | Electrochemical surface area (ECSA) measurement. Cyclic voltammograms with various scan rates for $\text{NH}_4\text{Cl-Cu}_2\text{O}$ (A), $\text{Cu}_\text{P}/\text{Cu}_2\text{O}$ (B) and $\text{Cu}_\text{I}/\text{Cu}_2\text{O}$ (C). (D) Current due to double-layer charging plotted against cyclic voltammetry scan rate.

Supplementary Figure 30 | FE of products derived from Cl-Cu₂O with different particle size: (A) ~30 nm, (B) ~150 nm, (C) ~500 nm, (D) ~1 μm in 1 M KOH electrolyte.

Supplementary Figure 31 | FE of products in 0.1 (A), 0.5 (B), 1 (C) M KHCO₃, 2 M KOH (D) and 3 M KCl (E) electrolyte, and (F) statistical bar chart of C₂₊ alcohols and ethylene for Cu₁/Cu₂O.

[2]. Carbon based gas diffusion layers (GDL) are used for coating the prepared catalysts. How did the authors mitigate electrolyte flooding at 200 mA/cm² current density? This is especially a common issue with KOH electrolyte and it's surprising that information on this aspect is not provided.

Response: We thank the reviewer for the professional question about electrolyte flooding, which is a very important issue using carbon based gas diffusion electrode. And, the commercial Sigracet 29BC gas diffusion layers are used for electrochemical tests in this manuscript, where a phenomenon of electrolyte flooding is also existent during a electrolysis process. However, the quantification of the relevant electrochemical properties is based on the volume of the electrolyte after electrolysis, therefore the product errors resulted from electrolyte flooding are negligible. It is a pity that no good strategy is provided for mitigating electrolyte flooding in this manuscript, and we will pay more attention for the study of electrolyte flooding in future study. The related modifications are marked red in the revised manuscript.

We have followed some studies about electrolyte flooding. For example, Rufford and co-workers have mitigated electrode flooding at high current densities using a vacuum-assisted

infiltration method to embed 200–400 nm-sized polytetrafluoro-ethylene (PTFE) particles at the interface of the microporous layer (MPL) and carbon cloth in a commercial GDL (ACS Energy Lett. 2022, 7, 2884–2892). In CO₂ electrolysis to CO over a silver nanoparticle catalyst on the GDL, the PTFE-embedded GDL not only just exhibited less than 10% of the electrolyte seepage rates observed in untreated GDLs at a current density of 300 mA cm⁻² but also expanded the electrochemical active area across the testing conditions. The PTFE-embedded GDL also maintained a Faradaic efficiency for CO₂ electrolysis to CO above 80% for more than 100 h at 100 mA·cm⁻², which was a 50-fold improvement in the stable operation time of the electrolyzer. In addition, unstable mass transport overpotential will impede higher current density operation (>200 mA cm⁻²) owing to gas evolution, preventing wide-scale commercialization. Here, Bazylak group identifies a real-time correlation between the electrolyte layer gas content and the cathode potential in an operating flow cell via concurrent galvanostatic operation and subsecond X-ray synchrotron imaging, whereby gas accumulation directly corresponds to increasing cathode overpotentials and gas removal corresponds to decreasing cathode overpotentials (ACS Sustainable Chem. Eng. 2021, 9, 5570–5579). And, it suggests that a 5% decrease in gas volume near the interface of the cathode gas diffusion electrode (GDE) and the electrolyte layer corresponds to a 12% decrease in the cathode overpotential at a current density of 125 mA cm⁻². Moreover, gas saturation becomes more stable at high current densities (>175 mA cm⁻²) due to more frequent gas removal, consequently stabilizing cell performance. Moreover, other articles give some references about electrolyte flooding, such as the acetone (acetone-based perfluorinated sulfonic acid (PFSA) ionomer Aquivion)-catalysts layers (ACS Appl. Mater. Interfaces 2023, 15, 52461–52472), the catalyst with cross-linked poly-diallyldimethylammonium chloride (Nat. Commun. 2023, 14, 5640), a hydrophobicity graded GDL (Appl. Catal. B Environ. 2023, 15, 122597) and so on.

Figure. Infiltration of polytetrafluoroethylene (PTFE) particles in a commercial gas diffusion layer (GDL). (a) Schematic of the vacuum-assisted infiltration method. (b) Energy-dispersive X-ray spectroscopy (EDS) mapping of fluorine in the cross section of the commercial GDL (F mass% = 4.09%) and (c) in the GDL after 30% P/C infiltration (F mass% = 10.41%). (d) Scanning electron microscope (SEM) image of the surface of the microporous layer (MPL) in the GDL after 30% P/C infiltration. (e) Gravimetric loading of the PTFE and carbon black on GDLs infiltrated with suspensions of different PTFE concentrations. (f) Pore size distribution of the 0% P/C and 30% P/C GDLs obtained from mercury intrusion porosimetry. The particle size of PTFE in the aqueous suspension (200–400 nm) is between the pore sizes of carbon cloth and MPL (ACS Energy Lett. 2022, 7, 9, 2884–2892).

[3]. Besides, no information on the type of carbon GDL used and PTFE content are provided. In addition, details of the flow cell design are missing.

Author response: We appreciate the reviewer for the professional question. I'm so sorry that the information about the type of carbon GDL, PTFE content and design of flow cell are absent in this manuscript, which have been updated in revised manuscript. Gas diffusion layers typically consist

of a bilayer structure consisting of a macro-porous backing material (PAN-based carbon fibers) and a micro-porous, carbon-based layer (MPL). Finishing of GDL comprises hydrophobic treatment of the substrate with PTFE and coating with a microporous layer (MPL). The commercial Sigracet 29BC gas diffusion layers are used with the standard microporous layer based on 77% carbon black and 23% PTFE. The related modifications are marked red in the revised manuscript.

Figure. Structure of gas diffusion layers.

Supplementary Figure 19 | Scheme of the three-electrode flow cell. GDL: gas diffusion layers. PMMA: polymethylmethacrylate.

[4]. The authors state that Ni foam was used as counter electrode. For the 50 hr test, what was the counter electrode or anode used? The pH of anolyte is well known to drop due to CO₂ crossover, where Ni/NiOOH is known to get corroded due to nickel carbonate formation.

Author response: We thank the reviewer for pointing out our mistake. For the 50 hr test, Ni foam was also used as the counter electrode, which indeed was corroded after electrolysis. According to the reviewer's suggestion, we have replaced anodic Ni foam with IrO₂/Ti mesh with IrO₂ loading of 1 mg/cm² as the counter electrode for studying the stability of cathode, which was shown in Fig. 4H. The related modifications are marked red in the revised manuscript.

Fig. 4 | Electrochemical performance of CO₂ reduction: (A) LSV curves acquired in flow cell using 1 M KOH as electrolyte. (B and C) Faradic efficiencies of products at various j_{total} using Cu_P/Cu₂O (B) and Cu_L/Cu₂O (C) catalysts. (D) C₂₊ and C₂₊ alcohols partial current density of Cu_P/Cu₂O and Cu_L/Cu₂O catalysts. (E and F) Energy efficiencies (E) and production rate (F) of C₂₊ alcohols on Cu_P/Cu₂O and Cu_L/Cu₂O catalysts. (G) Summary of EE_{C₂₊ alcohols} versus FE_{C₂₊ alcohols} of this work with other catalysts in flow cell. (H) Electrochemical stability test at the j_{total} of 200 mA cm⁻² using the Cu_L/Cu₂O catalysts.

[5]. The explanation on Matt-Schotky catalyst design is quite confusing and lacks clarity in a few places. How does the chlorine doped cuprous oxide prepared catalyst generate a mix of Cu⁺ and Cu(0) oxidation states during CO₂RR for 50 hrs? This is a big challenge with Cu based catalysts as the oxidation state is well known to reduce to 0 at these negative potentials. Does this mean techniques such as pulsed electrolysis on Cu based catalysts to regenerate Cu⁺ species are not essential compared to this catalyst design ?

Author response: We thank the reviewer for the good suggestion. It is a big challenge for avoiding

the metallization of Cu-based catalysts at a negative potential. Pulse methods present scientific and technological opportunities for the formation of copper oxide. P-eCO₂R has emerged as a simple and responsive knob to increase electrolyzer durability and improve product selectivity.

As shown in Figure below, a Pourbaix diagram provides a useful description for the dynamic changes in the electrode composition. Depending on the combination of applied potential and pH, copper can exist in a variety of oxidation states.

Figure. Pourbaix diagram. Identification of thermodynamically stable copper species at a given pH and potential (V versus SHE) at 25°C and $[\text{Cu}(\text{aq})]_{\text{tot.}} = 10^{-6}$ mol/kg. Dotted lines indicate water stability window and red arrows demonstrate possible phase change. Beverskog and Puigdomenech (J. Electrochem. Soc. 144, 3476–3483).

P-eCO₂R often involves anodic potentials for which the formation of oxide species is thermodynamically expected, which provides a very important strategy for observing the effects of copper oxides on CO₂RR. For example, Arán-Ais et al. used vacuum-transfer Auger electron spectroscopy to examine the nature of Cu during the p-eCO₂R reaction (Nat. Energy 2020, 5, 317–325). Their experiments focused on pulsing on single crystal Cu(100) ($E_c = -1$ V, $0 \leq E_a \leq 0.8$ V versus RHE, $t_a = t_c = 1$ s), where they uncovered that Cu⁺ sites form during the anodic pulse proportional to E_a , and that these sites persist during the cathodic pulse. When $E_a = 0.4$ V, they measured 14% of surface species to be Cu₂O, and 4% remained during cathodic pulse at -1 V

versus RHE. When testing a Cu(100) electrode with the same defect density but different Cu⁺ content. Besides, Lin et al., highlights the importance of Cu⁺ species in the ethanol formation (Nat. Commun. 2020, 11, 3525), demonstrating that a pulsed-potential program ($E_a = 0.5$ V, $-1.15 \leq E_c \leq -0.7$ V versus RHE, $t_a = t_c = 10$ s) can maintain the existence of Cu⁺ species throughout electrolysis using *operando* time-resolved XAS. Jeon et al. also imply that Cu⁺ and Cu⁰ sites coexist on the catalyst surface during pulsed electrolysis (J. Am. Chem. Soc. 2021, 143, 7578–758). A recent study by Tang et al. offers some additional insight into discrepancies observed around the effects of Cu⁺ and which C₂ product it enhances (ACS Appl. Mater. Interfaces 2021, 13, 14050–14055). Conducting temperature-controlled experiments between 5°C–25°C, they found that ethylene is favored at higher temperatures, while ethanol is favored at lower temperatures. These observations coincided with observations of a larger system RC time constant with decreased temperature, where the response time is the time in which the initial sharp current spike decays to a stable current after a potential step. Using CV scans and XRD to rule out changes in roughness and faceting, they interpret these results as lower temperatures effectively stabilizing Cu⁺ species (by slowing the reduction kinetics of Cu⁺ to Cu⁰) during the cathodic pulse which enhances ethanol selectivity. In addition, to deeply understand the selectivity and activity issues of the copper catalysts for the multi-carbon products, Yang et al. systematically prepared the F-stabilized Cu(I)/Cu(0) IF model catalyst series (P-Cu_x/Cu₂OF) elaborately constructed by a pulsed potential conversion process (Chem 2023, 10, 1–23), which shown in Figure below. Compared with only Cu(0) structure (P-Cu), the purposely designed P-Cu_x/Cu₂OF enables high activity and selectivity toward ethanol generation. By combining *in situ* and *ex situ* characterizations and DFT calculations, they have also unveiled different electroactivities at different Cu sites in the Cu(I)/Cu(0) model catalysts toward the C₂₊ products, dictated by the CO₂/CO adsorption preferences on those Cu sites, close to or away from the IF region.

Figure. Pulsed electrochemical generation of the catalysts (Chem 2023, 10, 1–23). (A) The atomic arrangements of precursors Cu(OH)F (bottom) and Cu(OH)₂ (top). (B) Schematic illustration of the pulsed electrochemical process and setup for P-Cu_x/Cu₂OF preparation from the precursor Cu(OH)F. (C) In situ monitoring of mass change in the Cu(OH)F to P-Cu_{1.65}/Cu₂OF transformation process with an electrochemical quartz crystal microbalance (EQCM). The mass signal arises from cyclic voltammetry scans sitting on applied square-wave voltage pulses. The homologous current response of the CV scans is also shown. (D) A representative enlarged portion of the dynamically balanced mass change profile from (C) in stage II and the corresponding current change profile recorded for the pulsed electrochemical formation process of P-Cu_{1.65}/Cu₂OF. (E) Schematic illustration of the anodic pulse for Cu_xO formation (left) and the cathodic pulse for Cu⁺ reduction (right) for the precursor transformation en route to P-Cu_{1.65}/Cu₂OF in stage II.

In details, Kimura et al. found that with 50 ms pulses on polycrystalline copper, the oxidation limit can be momentarily surpassed (ACS Catal 2020, 10, 8632–8639). Even though the electrode exists above the oxidation potential where Cu₂O can thermodynamically form, oxidation is kinetically limited such that only Cu⁰ exists throughout the 100 ms period (E_a = 0.6 V, E_c = -1.0 V versus RHE). In addition to avoiding copper oxidation by using millisecond pulses, copper oxidation can also be avoided during longer pulses by choosing a less anodic upper potential. In

experiments with longer pulse durations (5–60 s) where the anodic pulse potential is far below the equilibrium oxidation limit ($E_a = -0.8$ V versus RHE), enhanced selectivity toward CO₂R is still observed even though no oxides form (ACS Catal 2020, 10, 12403–12413). Taken together, these results imply that while Cu⁺ sites play a role in longer pulse programs with sufficiently anodic upper potentials, it does not explain the p-eCO₂R behavior at shorter pulses or when the upper potential is below the oxidation potential.

Thus, the controlled synthesis with pulsed potentials inspires me to insight the exquisite between the different valence Cu species in the Cu-based catalyst and p-eCO₂R technology.

And, there may be some origins for the stable oxidation state based on the structural features of Cu_L/Cu₂O in our manuscript, which present in a follow-up response.

The origins of Cu⁺ state in our manuscript can be summarized as follows:

(1) strain strengthening effect of residual Cl. The halogen ions play an important role in constructing Cu-based nanostructures and stabilizing cation Cu species (Nano Lett. 2019, 19, 3925–3932; Chem Electro Chem 2016, 3, 1012–1019; Angew. Chem. Int. Ed. 2022, 61, e202116706). On the other hand, halogen ions in electrolyte that strongly adsorb on the surface of Cu catalysts could influence the surface charge property as well as the binding strength of certain intermediates (Catal. Commun. 2018, 114, 109–113; J. Electrochem. Soc. 2016, 163, F477–F484). For example, the addition of I⁻ ions into the CsHCO₃ electrolyte was proposed to contribute to the stabilization of Cu⁺ species under CO₂RR reaction by adsorbed iodine ions (ACS Catal. 2018, 8, 10012–10020). Similarly, the use of KCl electrolyte was also demonstrated to induce a biphasic Cu₂O-Cu catalyst and preserve Cu⁺ species under CO₂RR (Angew. Chem. Int. Ed. 2015, 54, 14701–14705).

(2) the existence of coordination defects and oxygen vacancies for increase the adsorption of intermediates rather than proton bonding with O atoms of catalyst. For example, Li and co-workers prepared three Cu-based electrocatalysts with different oxidation states to study the valence state–activity relationship (J. Am. Chem. Soc. 2023, 145, 26133–26143). And, they suggest that the abundant oxygen vacancies can stabilize the oxidation state of Cu to improve the selectivity for C₂₊ products in CO₂RR. The electrocatalytic reduction process of Cu₂O to Cu in an aqueous solution involves the initial adsorption of H from H₂O onto the oxygen in Cu₂O. The energy

released during the formation of bonds by oxygen-adsorbed water molecules was -2.06 eV, while the adsorption of the *COOH group at the vacancy releases an energy of -2.66 eV. The higher energy release indicates a more spontaneous reaction process, which suggests that oxygen vacancies regulate the electronic structure around Cu active sites to avoid their reduction. Besides, Zeng group successfully synthesized ZnO nanosheets with abundant oxygen vacancy, which restrained the self-reduction of ZnO nanosheets to the metallic state and improved the activity toward CO₂ electroreduction because of the faster charge transfer (Angew Chem Int Ed. 2018, 57, 6054-6059). Zheng group reported a partially reduced CuO_x nanodendrites with rich oxygen vacancies surface, which served as excellent Lewis base sites for enhanced CO₂ adsorption and investigated the significance of oxygen vacancies on the stability of copper oxides (Small Methods 2019, 3, 1800449).

(3) strong electronic interactions between carbon intermediate and Cu⁺. For instance, the electron configuration of CO is $(1\sigma^2 2\sigma^2 3\sigma^2 4\sigma^2 1\pi^4 5\sigma^2)$, in which the HOMO is the 5σ orbital and the LUMO is the $2\pi^*$ orbital. When *CO bonds to the Cu surface, the 5σ and $2\pi^*$ orbitals of *CO are each split into bonding and anti-bonding orbitals due to hybridization with the d-states of Cu. In this context, the 5σ orbital of *CO will donate electrons to the d-band of Cu to form a 5σ -d donation, and the electrons in the d-band of Cu will be transported to $2\pi^*$ orbitals of *CO to form a d- $2\pi^*$ backdonation (J. Phys. Chem., 1964, 68, 2772–2777). The interaction between the d orbital of Cu and the frontier orbital of CO results in the formation of the bond. The electron back-donation weakens the internal connections between C and O because the $2\pi^*$ orbital is vacant and antibonding. Depending on the level of electron back-donation, the surface of Cu will adsorb CO molecules to varying degrees (ACS Catal. 2023, 13, 9222–9233; J. Phys.: Energy 2021, 3, 022001). Li group have shown differences of strong electronic interactions between carbon intermediate and Cu derived from Cu-NS, Cu₂O-NS and CuO-NS. As shown in Figure below, some of the electrons on Cu of Cu-NS are transferred to form an oxidized state, so that they can adsorb CO and OCCHO. Unlike Cu-NS, Cu on the surfaces of Cu₂O-NS and CuO-NS is already in an oxidized state, and Cu as an electron acceptor can effectively reduce the σ repulsion of *CO. The big aggregated electron clouds can be observed on the Cu–C bond, when CO is adsorbed (Figure i,j). At the same time, because CuO-NS has too strong of a binding energy for CO, the number of electrons transferred during C–C coupling is significantly higher than that required for Cu₂O-NS

to realize C–C coupling (Figure 1,m), which proved that Cu^+ was the optimal oxidation state for adsorbing CO, further facilitating the C–C coupling (J. Am. Chem. Soc. 2023, 145, 26133–26143). And, Li and co-workers suggest that CO_2RR promotes the preservation of Cu^+ species on catalyst surface (ACS Nano 2023, 17, 12884; Adv. Funct. Mater. 2023, 2310913). A significant $\ast\text{CO}$ coverage exists on the $\text{Cu}_2\text{O}/\text{CuO}$ surface during CO_2RR , which plays a crucial role in protecting and stabilizing the Cu^+ species. This effect stabilizes the active Cu^+ species during CO_2 electroreduction, which bind to Cu^+ sites to cover the local catalyst surface.

Figure. Charge density difference of CO and OCCHO intermediate adsorption on (h,k) Cu-NS, (i,l) Cu_2O -NS, and (j,m) CuO-NS (J. Am. Chem. Soc. 2023, 145, 26133–26143).

[6]. It is well known that sub-surface oxides are not stable under negative potentials. How does this catalyst (Cu then generate stable oxide species during CO_2RR ? Clarity on what the oxidation state of Cu is during CO_2RR but must be explained.

Author response: We thank the reviewer for the professional questions. In this manuscript, Operando SR-FTIR and Raman measurements were updated in Fig. 5A~B and Supplementary Figure 32~33. The electrochemical reconstruction of precursors are inevitable under the harsh CO_2 reduction conditions, but the $\text{Cu}_\text{I}/\text{Cu}_2\text{O}$ catalyst shows protected Cu-O bonds proofed from the Raman signals assigned at 224, 425, 522 and 626 cm^{-1} . The related modifications are marked red in the revised manuscript.

Supplementary Figure 33 | (A and B) Operando SR-FTIR and (C and D) Raman measurements with potentials from -0.2 ~ -1.1 V and time during 20 min at -0.6 V vs. RHE for $\text{Cu}_L/\text{Cu}_2\text{O}$.

Supplementary Figure 34 | (A) Operando Raman measurements with various potentials from -0.2 ~ -1.1 V and time during 20 min at -0.6 V vs. RHE of $\text{Cu}_P/\text{Cu}_2\text{O}$.

The Cu_2O and $\text{Cl-Cu}_2\text{O}$ samples exhibit characteristic signals at 224, 425, 522 and 626 cm^{-1} attributed to the $2\Gamma_{12}^-$, $4\Gamma_{12}^-$, Γ_{25}^+ , and $\Gamma_{12}^- + \Gamma_{25}^+$ phonon modes, respectively (Solid State Commun. 1969, 7, 815–818; Phys. Rev. B 1975, 12, 1377–1394). These Raman signals for the Cu_2O precursor disappear gradually in the potential range from -0.2 to -0.7 V vs RHE,

corresponding to the full Cu₂O reduction to metallic Cu. In sharp contrast, the Cl-Cu₂O retains the characteristic Raman modes ($2\Gamma_{12}^-$ and $\Gamma_{12}^- + \Gamma_{25}^+$ mode) until the potential > -0.7 V, suggesting that the low-coordinated Cu_L/Cu₂O can protect Cu⁺ species against reduction owing to the combination of environmental and structural factors. In addition, Raman measurement displays a prominent emerged peak at 2000~2100 cm⁻¹, which corresponds to the ν CO vibrational mode of *CO. Overall, Cu_L/Cu₂O catalyst has stronger adsorption capacity for *CO attributed to the restricted rotation and stretching vibration peaks of adsorbed *CO and Cu-CO bond around 310 and 358 cm⁻¹, respectively. In particular, the *CO coverage is closely related to the intensity ratio of the ν (Cu-CO) to the ν (*CO). Obviously, the intensity ratio of the ν (Cu-CO) to the ν (*CO) of Cu_L/Cu₂O is greater than Cu_L/Cu₂O, indicating that *CO coverage is higher and beneficial to the oxidation state of Cu⁺.

In briefly, there are three reasons for oxidation state of Cu_L/Cu₂O in our manuscript, which are enumerated in Question [5] above, listing briefly here: (1) strain strengthening effect of residual Cl. The halogen ions play an important role in constructing Cu-based nanostructures and stabilizing cation Cu species (Nano Lett. 2019, 19, 3925–3932; Chem Electro Chem 2016, 3, 1012–1019; Angew. Chem. Int. Ed. 2022, 61, e202116706). For example, the addition of I⁻ ions into the CsHCO₃ electrolyte was proposed to contribute to the stabilization of Cu⁺ species under CO₂RR reaction by adsorbed iodine ions (ACS Catal. 2018, 8, 10012–10020). (2) the existence of coordination defects and oxygen vacancies for increase the adsorption of intermediates rather than proton bonding with O atoms of catalyst. The abundant oxygen vacancies can stabilize the oxidation state of Cu to improve the selectivity for C₂₊ products in CO₂RR (J. Am. Chem. Soc. 2023, 145, 26133–26143)). The electrocatalytic reduction process of Cu₂O to Cu in an aqueous solution involves the initial adsorption of H from H₂O onto the oxygen in Cu₂O. The energy released during the formation of bonds by oxygen-adsorbed water molecules was -2.06 eV, while the adsorption of the *COOH group at the vacancy releases an energy of -2.66 eV. The higher energy release indicates a more spontaneous reaction process, which suggests that oxygen vacancies regulate the electronic structure around Cu active sites to avoid their reduction. (3) strong electronic interactions between carbon intermediate and Cu⁺. For instance, when *CO bonds to the Cu surface, the 5 σ and 2 π^* orbitals of *CO are each split into bonding and anti-

bonding orbitals due to hybridization with the d-states of Cu. In this context, the 5σ orbital of *CO will donate electrons to the d-band of Cu to form a 5σ -d donation, and the electrons in the d-band of Cu will be transported to $2\pi^*$ orbitals of *CO to form a d- $2\pi^*$ backdonation (J. Phys. Chem., 1964, 68, 2772–2777). Depending on the level of electron back-donation, the surface of Cu will adsorb CO molecules to varying degrees (ACS Catal. 2023, 13, 9222–9233; J. Phys.: Energy 2021, 3, 022001). As a result of these factors above, Cu_L/Cu_2O maintains the oxidation state of $Cu^{\sigma+}$ (Supplementary Figure 17).

Supplementary Figure 17 | Chemical valence of Cu atom in Cu foil, Cu_2O , CuO and Cu_L/Cu_2O .

[7]. The authors state, “the fast electron transfer from Cu to Cu^+ at the enhanced rectifying interface can induce lattice shrinkage and inhibit oxygen atoms loss”. In a few lines subsequent to it, it is stated, “Third, abundant defects originated from oxygen vacancies and Cl doping supply rich free electron and landing sites for adsorbed intermediates “. This is quite confusing. Does it inhibit oxygen atoms loss (that is retain oxygen) or create oxygen vacancies?

Author response: We thank the reviewer for the professional questions. I apologize for confusing readers owing to some conflicting statements in the manuscript. Oxygen vacancies of Cu_L/Cu_2O are derivative of electrochemically reconstructing Cl- Cu_2O , where the residual oxygen atoms stabilize the oxidation states of Cu_L/Cu_2O . In other words, oxygen vacancy is a result of

reconstruction of catalyst, after that, the residual oxygen atoms are responsible for the stability of oxidation state in $\text{Cu}_L/\text{Cu}_2\text{O}$. The compressive strain attributed to the lattice vibration can restrict residual oxygen migration for protecting the oxidation state of the desired Cu^+ sites (Adv. Funct. Mater. 2016, 26, 1564–1570. ACS Nano 2013, 7, 3276–3286). Owing to the interaction between electrons and lattice vibration, when the lattice atomic vibration is offset (scattering takes place), there is an exchange of momentum and energy between electrons and lattice vibration, resulting in coupling between electrons and phonons. For example, the interphase strain between Cu and oxidized Cu will be enhanced by lattice shrinkage, which will further inhibit oxygen loss (J. Am. Chem. Soc. 2023, 145, 8714–8725). The nanocrystal-line Cu embedded in the amorphous CuO matrix creates notable compressive strain at the interface regions. Meanwhile, electron transfer from Cu to CuO at the interphase yields Cu^+ species, which are well stabilized since lattice shrinkage can suppress oxygen loss, protecting the chemical state of interfacial Cu. Thus, an improvement of electronic conduction because of the fast electron transfer from Cu to Cu^+ at the enhanced rectifying interface in $\text{Cu}_L/\text{Cu}_2\text{O}$ can result in a stronger lattice vibration and inhibit residual oxygen atoms loss for protecting the oxidation state of Cu^+ (ACS Energy Lett. 2023, 8, 4540–4546). The related modifications are marked red in the revised manuscript.

[8]. For the preparation of catalyst, the nanoparticles are dispersed ethanol as the solvent and heated at 80 deg. After this, they also use ethanol to wash it multiple times. Are the authors sure that the ethanol produced from CO₂RR do not come from this probable residual ethanol?

Author response: We thank the reviewer for the good questions. And, to verify the sources of ethanol product in CER, we have synthesized Cu_2O and Cl- Cu_2O samples without use of ethanol and collect some new dates from repeated electrolysis experiments. As shown in Figure below, the selectivity of C_{2+} alcohols has been maintained a stable value without substantial decline, which proof the validity of dates. And, the relative experiments are also avoided to use ethanol in the revised manuscript. The related modifications are marked red in the revised manuscript.

Figure. (A, B) SEM images and (C, D) FE of products derived from $\text{Cu}_I/\text{Cu}_2\text{O}$ (A and C) and $\text{Cu}_L/\text{Cu}_2\text{O}$ (B and D) without using ethanol for the synthesis of precursors.

Minor Comments

1. It might be better to report the partial current densities of ethanol instead of the Faradaic Efficiency, especially in Table S5.

Author response: We thank the reviewer for the good questions. We have updated the date of Supplementary Table 5. The related modifications are marked red in the revised manuscript. Supplementary Table 5. A comparison of C_{2+} alcohols on various electrocatalysts.

Catalysts	Electrolyzer	Electrolytes	Potential (V vs. RHE)	j_{total} ($\text{mA}\cdot\text{cm}^{-2}$)	j_{ethanol} ($\text{mA}\cdot\text{cm}^{-2}$)	$\text{EE}_{\text{alcohols}}$	Reference
$\text{Cu}_I/\text{Cu}_2\text{O}$	Flow cell	1 M KOH	-0.66	-200	-113	39.32	This work
K-F-Cu-CO ₂	Flow cell	1 M KOH	-0.53	-800	-423	36.45	Adv. Mater. 2022, 34, 2204476
BaO/Cu	Flow cell	1 M KOH	-0.75	-400	-204	35.06	Nat. Catal. 2022, 5, 1081–1088
Cu-DS	H-cell	0.1 M KHCO ₃	-1.08	-32	-16	34.96	Joule 2021, 5, 429–440
	Flow cell	1 M KOH	-0.95	-200	-50	34.97	

R-Cu-C	Flow cell	1 M KOH	-0.8	-47.4	15.4	33.18	Chem Catal. 2023, 3, 100512
N-C/Cu	Flow cell	1 M KOH	-0.68	-300	-156	31.86	Nat. Energy 2020, 5, 478–486
NGQ/Cu-nr	Flow cell	1 M KOH	-0.9	-281.2	-126.5	27.8	Angew. Chem. Int. Ed. 2020, 59, 16459–16464
CuZn alloy	Flow cell	1 M KOH	-0.68	-200	-84	27.43	Angew. Chem. Int. Ed. 2019, 58, 15036–15040
wr-Cu	Flow cell	1 M KOH	-0.91	-800	-328	26.43	Chem. Sci. 2023, 14, 310–316
FeTPP[Cl]/Cu	Flow cell	1 M KHCO ₃	-0.82	-302	-124	24.56	Nat. Catal. 2020, 3, 75–82
Hex-2Cu-O	H-cell	0.1 M KHCO ₃	-1.2	-8.5	-2.76	23.76	Nat. Commun. 2022, 13, 5122
	Flow cell	1 M KOH	-0.66	-200	-26	23.25	
Cu-CuI	Flow cell	1 M KOH	-1	-900	-261	18.35	Angew. Chem. Int. Ed. 2021, 60, 14329–14333
Cu(OH) ₂ -D	Flow cell	1 M KOH	-0.54	-250	-42.5	17.99	Angew. Chem. Int. Ed. 2021, 60, 4879–4885
Li _{2-x} CuO ₂ -10	Flow cell	1 M KOH	-0.85	-220	-59.4	16.21	Small 2022, 18, 2106433
F-Cu	Flow cell	2.5 M KOH	-0.54	-400	-68	12.86	Nat. Catal. 2020, 3, 478–487
Cu ₂ S-Cu-V	Flow cell	1 M KOH	-0.95	-400	-92	9.45	Nat. Catal. 2018, 1, 421–428
Cu-CIPH	Flow cell	7 M KOH	-0.91	-1170	269.1	7.3	Science 2020, 367, 661–666
Cu/NPC-800	H-cell	0.2 M KHCO ₃	-1.05	-12.13	8.2	36.61	Green Chem. 2020, 22, 71–84
Cu ₂ O-ZnO	H-cell	0.5 M KHCO ₃	-0.9	-	-	31.47	J. Power Sources, 2023, 556, 232468
Cu ₃ Ag ₁	H-cell	0.5 M KHCO ₃	-0.95	-39.7	-17.2	27.64	Adv. Energy Mater. 2020, 10, 2001987
CeO ₂ -Cu	H-cell	0.1 M KHCO ₃	-0.6	45.3	13.14	20.45	ACS Materials Lett. 2022, 4, 1999–2008
Cu_I	H-cell	0.1 M KHCO ₃	-0.9	-38	-10.5	19.76	Angew. Chem. Int. Ed. 2019, 58, 17047–17053
CuO-ZnO ₁₀	H-cell	0.5 M KHCO ₃	-0.8	-5	-3.6	16.26	Electrochimica Acta 2021, 392, 138988
od-Pd ₉ Cu ₉₁	H-cell	0.5 M KHCO ₃	-0.65	-8.4	-1.3	12.54	Green Chem. 2020, 22, 6497–6509

2. At some places, the units of current density are missing.

Author response: We thank the reviewer for the good questions. We have revised the mistakes of manuscript. The related modifications are marked red in the revised manuscript.

Thanks again for the reviewer's precious time and professional suggestions, which are very important as an insightful guidance for us to further improve the quality of our manuscript. We sincerely hope that our revisions according to the reviewer's suggestions can now meet the high standards of Nature communications. We really appreciate your precious time on reviewing our revised manuscript. and we hope that the appropriate revision will improve the article's quality and provide readers with a good experience. Best wish for you!

REVIEWER COMMENTS

Reviewer #1 (Remarks to the Author):

All my concerns have been addressed in the revised ms.

Reviewer #2 (Remarks to the Author):

The authors have addressed most of my concerns.

However, in the Response to Q1, the description "but the CuL/Cu₂O catalyst shows protected Cu-O bonds proofed from the Raman signals assigned at 224, 425, 522 and 626 cm⁻¹" is inaccurate, because it is evident that the peaks at 425 and 522 cm⁻¹ disappear at low potentials (Supplementary Figure 33c). So the operando Raman cannot confirm the presence of Cu⁺.

Furthermore, I would suggest that the authors address the questions directly. Some of the Responses contain too much redundant and irrelevant information, which hampers the review.

Reviewer #3 (Remarks to the Author):

The authors have now revised the MS with some new experiments and very detailed responses to reviewers' comments which is commendable and has improved the overall quality of the manuscript. The authors have also provided their explanation using other relevant papers published in the field. I accept the newly revised version of the MS provided the authors address these small changes in the MS:

1) The authors can state very briefly on how Cu⁺ species under cathodic potentials can still be preserved using their Mott-Schotky catalyst design with the references they used and how it is different from potentiodynamic techniques such as pulse electrolysis.

2) Second, the authors should briefly clarify in the main MS, whether this strategy and other catalyst designs like introduction of oxygen vacancies can show a stable operation at industrially relevant conditions (> 1000 hrs). All papers cited by the authors in their explanation only shows stable operation at higher current densities for 10-50 hrs. For >200 hrs of operation, a pulse electrolysis strategy may still be essential and the authors should discuss this briefly in the MS. This will provide a much more clarity for the CO₂ electrolysis community.

Response to the Reviewers' Comments

Reviewer #1: All my concerns have been addressed in the revised ms.

Author response: Thanks again for the reviewer's precious time and professional suggestions as an insightful guidance for us to further improve the quality of our manuscript. We really appreciate your precious time on reviewing our revised manuscript.

Reviewer #2: The authors have addressed most of my concerns.

Author response: Thanks again for the reviewer's precious time and professional suggestions. We sincerely hope that our revisions according to the reviewer's suggestions can now address the concerns of readers. We really appreciate your precious time on reviewing our revised manuscript.

[1] However, in the Response to Q1, the description "but the Cu_L/Cu₂O catalyst shows protected Cu-O bonds proofed from the Raman signals assigned at 224, 425, 522 and 626 cm⁻¹" is inaccurate, because it is evident that the peaks at 425 and 522 cm⁻¹ disappear at low potentials (Supplementary Figure 33c). So the operando Raman cannot confirm the presence of Cu⁺. Furthermore, I would suggest that the authors address the questions directly. Some of the Responses contain too much redundant and irrelevant information, which hampers the review.

Author response: Thanks for the reviewer's professional suggestions. First of all, we really appreciate the reviewer's valuable suggestion regarding the issue of two much redundant and irrelevant information during our previous response. In this version, according to your suggestion, we try our best to follow the rule of addressing the questions directly and avoiding redundant and irrelevant information. We also sincerely expect that our responses can address the reviewer's concerns.

The origin and conclusion of disappeared signals for Cu-O (425, 522 cm⁻¹) can be described as follows: 1) the intensity of Cu-O of Cu₂O would be weaker than that of OCP because the generation of Cu_L/Cu₂O mott-schottky catalyst is derived from the partial reduction of Cl-Cu₂O, resulting in the restricted Raman signals. 2) the signals of 425 and 522 cm⁻¹ are classified as the inconspicuous ones based on the Raman results of OCP in Supplementary Figure 33C, so these peaks are difficult to be recorded by operando Raman with the generation of Cu_L/Cu₂O mott-schottky catalyst and weakened intensity response for Cu-O signals. 3) the residual peaks also could prove the oxidation state of Cu (224, 626 cm⁻¹). For examples, Yu and workers have compared the different operando Raman spectra of multihollow, fragmental, and solid Cu₂O samples, where the multihollow Cu₂O retains the characteristic Raman modes (e.g., the 2Γ⁻₁₂ mode), suggesting that the confined geometry can protect Cu⁺ species against reduction (J. Am. Chem. Soc. 2020, 142, 6400–6408). In addition, Yan and workers have presented the stabilized Cu₂O nanoparticles decorated with hexagonal boron nitride (h-BN) nanosheets, which are proved

by operando Raman technology (Angew. Chem. Int. Ed. 2022, 61, e202205832). Where the characteristic peaks are retained in the spectrum of Cu₂O-BN, it indicates the stabilization of the copper valence state. Gewirth and workers study the correlation between the decreased oxide content on the Cu surface and increased presence of CO as well as increased activity for CO and C₂ production by using In situ surface enhanced Raman spectroscopy (SERS). During CO₂ reduction, only the T_{2g} vibrational mode remains on the surface (ACS Catal. 2020, 10, 672–682). Schultz and workers have investigated the oxidation state of Cu₂O using in-situ Raman spectra, where a weak peak assigned at ~620 cm⁻¹ is retained after electrolysis of 30 min (ACS Catal. 2023, 13, 1638–1648). So, the residual peaks of operando Raman can confirm the presence of Cu⁺.

Figure. Stabilizing Cu oxidation state via confined intermediates. Operando Raman spectra of multihollow (a), fragmental (b), and solid (c) Cu₂O as a function of reaction time at -0.61 V vs RHE (J. Am. Chem. Soc. 2020, 142, 6400–6408).

Figure. In situ Raman spectra of Cu₂O-BN as a function of reaction time at -0.4 V (Angew. Chem. Int. Ed. 2022, 61, e202205832).

Figure. Potential-dependent in situ SERS spectra obtained from (a) Cu-DAT, (b) CuAg-DAT, and (c) CuSn-DAT in a flow Raman cell with a 1 M KOH flow rate of 1 mL min⁻¹ and a CO₂ flow rate of 7 SCCM (ACS Catal. 2020, 10, 672–682).

Figure. Raman spectra of Ag/Cu₂O electrodes for CO₂ reduction carried out under 455 nm LED illumination and applying -0.4 V vs Ag/AgCl for 30 min (ACS Catal. 2023, 13, 1638–1648).

The origin of residual characteristic peaks assigned at 224 and 626 cm⁻¹:

The Raman peaks of Cl-Cu₂O precursor assigned at 224, 425, 522, and 626 cm⁻¹, which are attributed to the 2 Γ_{12}^- , 4 Γ_{12}^- , Γ_{25}^+ , and $\Gamma_{12}^- + \Gamma_{25}^+$ phonon modes. Group-theoretical analysis predicts that phonons with symmetry Γ_{12}^- can assist in the dipole excitation of the 1s exciton. Experimentally, it was found that the phonon-assisted absorption edge of Cu₂O was dominated by the contribution of the Γ_{12}^- phonon. Since a Γ_{12}^- phonon-assisted photoexcitation of the 1s exciton and radiative recombination of the exciton, the 2 Γ_{12}^- mode showed the strongest enhancement and

abruptly over the entire tuning range of potentials, where the absorption edge is derived from the photoexcitation of the 1s yellow exciton with emission of Γ^-_{12} phonons. In addition, the Raman signals of some silent modes will be enhanced due to nonstoichiometry or impurities. Thus, $\Gamma^-_{12} + \Gamma^+_{25}$ phonon mode also presents a recognizable signal value owing to the atoms defect of low-coordinated $\text{Cu}_L/\text{Cu}_2\text{O}$ (Phys. Rev. B 1975, 12, 1377–1394). The related modifications are marked red in the revised manuscript.

Reviewer #3: The authors have now revised the MS with some new experiments and very detailed responses to reviewers' comments which is commendable and has improved the overall quality of the manuscript. The authors have also provided their explanation using other relevant papers published in the field. I accept the newly revised version of the MS provided the authors address these small changes in the MS:

Author response: We thank the reviewer for the precious time on reviewing our manuscript. And, I'm so happy that our experiments and responses can address the concerns of reviewer and appreciate that reviewer can accept the revised version. And, We have revised the manuscript based on the new suggestions of reviewer and provided below:

[1] The authors can state very briefly on how Cu^+ species under cathodic potentials can still be preserved using their Mott-Schotky catalyst design with the references they used and how it is different from potentiodynamic techniques such as pulse electrolysis.

Author response: Thanks for the reviewer's good suggestions. We have revised the manuscript with a brief very for the difference between the Mott-Schotky catalyst design and potentiodynamic techniques. The revision is presented in "Introduction" described as "Therefore, the feasible strategies are demanded to develop the structural design of catalysts and electrolytic system for maintaining the oxidation state and increasing CO_2 performance of copper-based catalysts such as elemental doping, interface engineering, intermediate confinement, and pulse CO_2 electrolysis (P-e CO_2 R owing the simple and responsive knob to operate anodic potentials for the formation of oxide species of Cu), especially the Mott-Schottky catalyst because of the suppressed electron accumulation for protecting the Cu-O bonds and fast electron transfer with the built-in electric field". The related modifications are marked red in the revised manuscript.

[2] Second, the authors should briefly clarify in the main MS, whether this strategy and other catalyst designs like introduction of oxygen vacancies can show a stable operation at industrially relevant conditions (> 1000 hrs). All papers cited by the authors in their explanation only shows stable operation at higher current densities for 10-50 hrs. For >200 hrs of operation, a pulse electrolysis strategy may still be essential and the authors should discuss this briefly in the MS. This will provide a much more clarity for the CO_2 electrolysis community.

Author response: Thanks for the reviewer's the suggestions with highly strategic vision. We have revised the manuscript to clarify the importance of pulse electrolysis for a long-terms industrial production. The revision is presented in "Performance of CER in flow cell" described as "However, the pulse electrolysis is necessary to be mentioned for a long-terms industrial production, which may still be an essential strategy for CO₂ electrolysis community, especially the unique advantage in electrolysis stability (> 1000 h)". The related modifications are marked red in the revised manuscript.

Thanks again for the reviewers's precious time and professional suggestions, and we hope that the appropriate revision will improve the article's quality and provide readers with a good experience. Best wish for you!

REVIEWERS' COMMENTS

Reviewer #2 (Remarks to the Author):

All my concerns have been addressed.